# EVENT CAMERA OBJECT DETECTION AT ARBITRARY FREQUENCIES

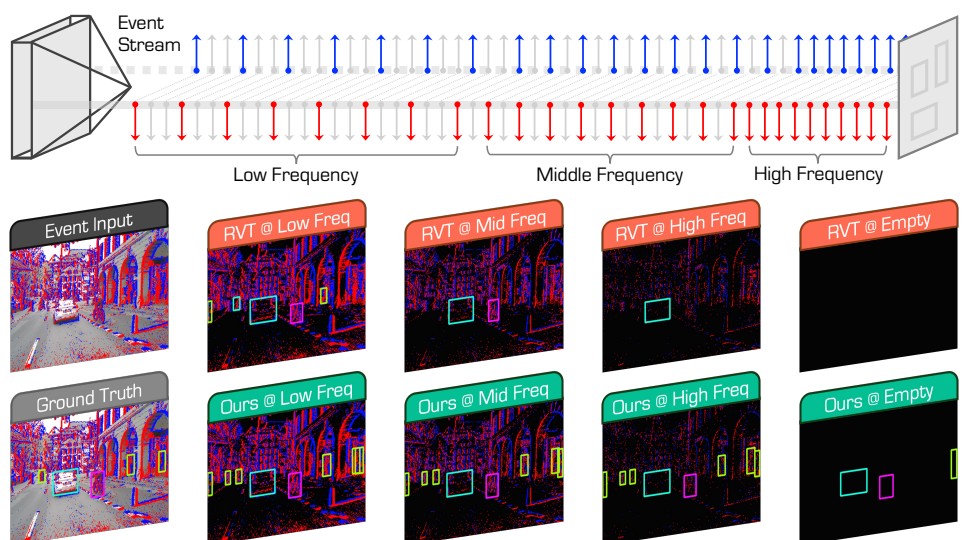

Figure 1: **Event camera detection at varying frequencies**. The performance of the classic RVT detector (Gehrig & Scaramuzza, 2023) drops significantly at higher event operational frequencies. Motivated by this, we propose FlexEvent, a robust and flexible detector that maintain high accuracy across a wide range of frequencies, ensuring strong adaptability in dynamic sensing environments.

## ABSTRACT

Event cameras offer unparalleled advantages for real-time perception in dynamic environments, thanks to their microsecond-level temporal resolution and asynchronous operation. Existing event-based object detection methods, however, are limited by fixed-frequency paradigms and fail to fully exploit the high-temporal resolution and adaptability of event cameras. To address these limitations, we propose FlexEvent, a novel event camera object detection framework that enables detection at arbitrary frequencies. FlexEvent consists of two key components: **FlexFuser**, an adaptive event-frame fusion module that integrates high-frequency event data with rich semantic information from RGB frames, and **FAL**, a frequency-adaptive learning mechanism that generates frequency-adjusted labels to enhance model generalization across varying operational frequencies. This combination allows FlexEvent to detect objects with high accuracy in both fast-moving and static scenarios, while adapting to dynamic environments. Extensive experiments on large-scale event camera datasets demonstrate that our approach surpasses state-of-the-art methods, achieving significant improvements in both standard and high-frequency settings. Notably, FlexEvent maintains robust performance when scaling from 20 Hz to 90 Hz and delivers accurate detection up to 180 Hz, proving its effectiveness in extreme conditions. Our framework sets a new benchmark for event-based object detection and paves the way for more adaptable, real-time vision systems. The code will be made publicly available to facilitate future research.

## 1 INTRODUCTION

Event cameras have garnered significant attention for their ability to capture dynamic scenes with microsecond-level temporal resolution (Gallego et al., 2022). Unlike conventional RGB cameras that

capture entire frames at fixed intervals, event cameras operate asynchronously, responding to changes in pixel intensity at each location (Zou et al., 2022). This low-latency operation reduces motion blur and enables highly energy-efficient sensing, making event cameras ideal for real-time applications such as autonomous driving, robotics, and surveillance (Steffen et al., 2019).

Despite their potential, existing event camera object detection methods often fail to fully leverage the high-frequency temporal information captured by these cameras (Cordone et al., 2022; Gehrig & Scaramuzza, 2022; Jeziorek et al., 2023). Most approaches align event data with the lower frequency of RGB cameras by adopting a fixed time interval between event streams and frame-based annotations (Perot et al., 2020; Gehrig et al., 2021b). While this strategy simplifies data processing, it inevitably overlooks the rich temporal details embedded in high-frequency event streams, limiting its adaptability to dynamic environments where temporal changes occur at varying rates (Perot et al., 2020). Given that human annotations are often synchronized with slower frame rates, current detection models miss valuable information from high-frequency event data, resulting in suboptimal performance when rapid object detection is required in dynamic environments (Messikommer et al., 2020; Schaefer et al., 2022).

To address these limitations, we introduce FlexEvent, a novel event camera object detection framework designed to tackle the challenging problem of object detection at varying operational frequencies. Our approach addresses the need for high-frequency detection in fast-changing environments, while adapting to different operational frequencies. We propose two key innovations: **(1) FlexFuser**, an adaptive event-frame fusion module, and **(2) FAL**, a frequency-adaptive learning mechanism.

**Flexible Event-Frame Fusion.** The first component, FlexFuser addresses the limitations of event data, which often lacks semantic and texture-rich information, especially at higher frequencies (Zhou et al., 2023), by synchronizing event data with frames and integrating the rich spatial and semantic information from frames with the high-temporal resolution of event streams. It enables high detection accuracy even in fast-moving environments. Furthermore, training on high-frequency event data is computationally expensive and impractical due to the significant human effort required to label such data. FlexFuser mitigates this by sampling event data at varying frequencies, aligning them with the normal frame rate during training, thus maintaining efficiency while preserving the high-frequency benefits at inference time.

**Frequency-Adaptive Learning.** The second component, FAL, enhances the generalization capability of event camera detectors across varying operational frequencies, by generating frequency-adaptive labels for the unlabeled high-frequency data. These labels allow the model to learn from high-frequency event streams without manual annotations, and iterative refinement through self-training ensures that the model remains robust across different motion dynamics and frequency settings. Together, these two components allow for accurate real-time detection in rapid scene changes and adapt to a wide range of operational frequencies, by leveraging the temporal richness of event data and the semantic detail of RGB frames.

Our extensive experiments validate the effectiveness of FlexEvent on multiple large-scale event camera datasets. Our approach consistently outperforms recent detectors across both standard and high-frequency settings. In particular, we achieve mAP **gains of 15.5%**, **9.4%**, and **10.3%** over previous best-performing detectors on the *DSEC-Det* (Gehrig & Scaramuzza, 2024), *DSEC-Detection* (Tomy et al., 2022), and *DSEC-MOD* (Zhou et al., 2023) datasets, respectively. Our model also **maintains 96.2% of its performance** when the operational frequency shifts from **20 Hz** to **90 Hz**, and delivers accurate detection at frequencies **as high as 180 Hz**, proving its robustness under extreme conditions.

In summary, our contributions are listed as follows:

▶ The FlexEvent framework is designed to tackle the challenging problem of event camera object detection at arbitrary frequencies, being one of the early attempts on this line of study.

▶ We propose FlexFuser, an adaptive event-frame fusion that leverages the strengths of both event and frame data, enabling efficient and accurate detection in dynamic environments.

▶ We introduce FAL, a frequency-adaptive learning mechanism that generates frequency-adjusted labels and improves generalization across a wide range of motion frequencies.

▶ We demonstrate that our approach achieves state-of-the-art performance in event-based object detection across large-scale datasets, particularly in high-frequency scenarios, validating its effectiveness and potential to handle safety-critical problems in the real world.

## 2 RELATED WORK

**Event Camera Object Detection.** Event-based detection methods can be broadly split into two approaches: GNNs/SNNs and dense feed-forward models. GNNs build dynamic spatio-temporal graphs by subsampling events (Gehrig & Scaramuzza, 2022; Sun & Ji, 2023; Messikommer et al., 2020; Schaefer et al., 2022), but they face challenges in propagating information over large spatio-temporal regions, especially for slow-moving objects. SNNs offer efficient sparse information transmission but are often hindered by their non-differentiable nature, complicating optimization processes (Cuadrado et al., 2023; Cordone et al., 2022; Zhang et al., 2022). Dense, feed-forward models represent the second approach. Initial methods using fixed temporal windows (Chen, 2018; Iacono et al., 2018; Jiang et al., 2019) struggled with slow-moving or stationary objects due to their limited capability to capture long-term temporal data. Subsequent advancements incorporated RNNs and transformers to enhance temporal modeling capabilities (Perot et al., 2020; Zubić et al., 2023; Li et al., 2022; Gehrig & Scaramuzza, 2023; Peng et al., 2024), but these models often still lack semantic richness and face difficulties in adapting to variable frequencies. EventDrop (Gu et al., 2021) and Shadow Mosaic (Peng et al., 2023) improve generalization using data augmentation techniques that introduce spatial and temporal manipulations to increase data diversity. However, they do not focus on high-frequency event data or fully leverage the rich temporal information of event streams.

**Event-Frame Multimodal Learning.** To overcome the limited texture in event streams, multimodal fusion techniques combining event-based and frame-based data have gained traction across tasks, such as deblurring (Sun et al., 2022a; Zhang et al., 2020), depth estimation (Gehrig et al., 2021a; Uddin et al., 2022), and tracking (Zhao et al., 2022; Gehrig et al., 2020). Earlier object detection approaches fused event and image data during post-processing (Li et al., 2019; Chen et al., 2019), but they lacked meaningful feature-level interaction. Recent works focus on deeper feature fusion (Tomy et al., 2022; Cao et al., 2022; 2021), with advanced methods introducing pixel-level spatial attention or temporal transformers for asynchronous processing (Zhou et al., 2023; Li et al., 2023; Gehrig & Scaramuzza, 2024; Cao et al., 2024). Some early attempts (Li et al., 2023; Gehrig et al., 2021a) explore combining events and frames through asynchronous multi-modal fusion, enabling inference at varying frequencies. However, they do not focus on high-frequency event data and fully leverage the temporally rich nature of event streams. Furthermore, these methods still face challenges in fully exploiting complementary strengths and addressing feature imbalance in event-frame detection. Unlike previous methods, FlexEvent employs a more comprehensive fusion framework that effectively combines high-temporal resolution event data with rich semantics from RGB frames, enabling robust object detection across varying frequencies while addressing feature imbalance.

**Label-Efficient Learning in Event Data.** Due to limited annotated datasets, label-efficient learning has become an important area for event-based vision. Several studies attempt to reconstruct images from event data (Rebecq et al., 2019; 2021; Stoffregen et al., 2020) or leverage knowledge distillation from pre-trained frame-based models (Wang et al., 2021; Sun et al., 2022b; Yang et al., 2023; Kong et al., 2024). Other approaches utilize pre-trained models or self-supervised losses (Klenk et al., 2022; Wu et al., 2023; Zhu et al., 2019). LEOD (Wu et al., 2024) pioneered object detection with limited labels but did not address high-frequency generalization. A recent state-space model (Zubić et al., 2024) adapts to varying frequencies without retraining but struggles with detecting static objects at high frequencies due to reliance solely on event data. In contrast, FlexEvent is specifically designed to adapt to varying event frequencies, ensuring consistent performance even in scenarios with limited labels, and effectively detecting both stationary and fast-moving objects.

## 3 FLEXEVENT: A FLEXIBLE EVENT OBJECT DETECTOR

In this section, we elaborate on the technical details of our FlexEvent framework. We start with the foundational concepts of event data and their representation in Sec. 3.1. We then introduce the **FlexFuser** module in Sec. 3.2, which adaptively fuses event and frame data to enhance detection across varying frequencies. Finally, we detail the frequency-adaptive learning (**FAL**) mechanism in

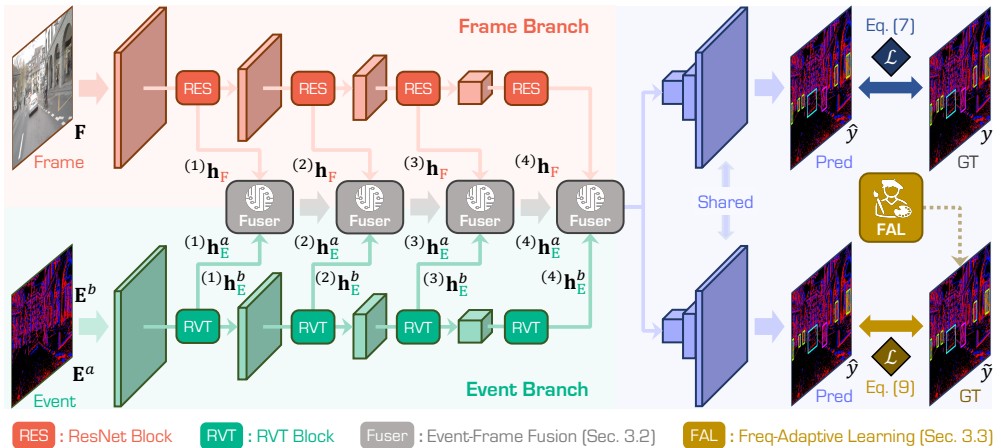

Figure 2: **Framework Overview.** The proposed FlexEvent consists of two branches: Event and Frame. The event branch captures high-temporal resolution data, while the frame branch leverages the rich semantic information from frames (*cf.* Sec. 3.1). These branches are fused dynamically through **FlexFuser**, allowing adaptive integration of event and frame data (*cf.* Sec. 3.2). Additionally, the frequency-adaptive learning (**FAL**) mechanism ensures robust detection performance across varying operational frequencies (*cf.* Sec. 3.3). Together, these components enable the model to handle diverse motion dynamics and maintain high detection accuracy in both low- and high-frequency scenarios.

Sec. 3.3, which enables our model to generalize effectively across diverse temporal conditions using self-training and adaptive label generation. The overall framework is illustrated in Fig.2.

## 3.1 PRELIMINARIES

**Event Processing.** Event cameras are bio-inspired vision sensors that capture changes in log intensity per pixel asynchronously, rather than capturing entire frames at fixed intervals. Formally, let $I(x, y, t)$ denote the log intensity at pixel coordinates $(x, y)$ and time $t$. An event $e$ is generated at $(x, y, t)$ whenever the change in log intensity $\Delta I$ exceeds a certain threshold $C$. Such a process can be modeled as:

$$\Delta I(x, y, t) = I(x, y, t) - I(x, y, t - \Delta t) \geq C . \tag{1}$$

Each event $e$ is a tuple $(x, y, t, p)$, where $(x, y)$ are the pixel coordinates, $t$ is the timestamp, and $p = \{-1, 1\}$ denotes the polarity of the event which indicates the direction of the intensity change.

To leverage event data with convolutional neural network layers, we preprocess events into a 4D tensor $E$ with dimensions representing the polarity, temporal discretization $T$, and spatial dimensions $(H, W)$. This representation involves mapping a setting of events $\mathcal{E}$ within time interval $[t_a, t_b)$ into:

$$E(p, \tau, x, y) = \sum_{e_k \in \mathcal{E}} \delta(p - p_k)\delta(x - x_k, y - y_k)\delta(\tau - \tau_k), \quad \tau_k = \left\lfloor \frac{t_k - t_a}{t_b - t_a} \cdot T \right\rfloor . \tag{2}$$

The tensor captures event activity in $T$ discrete time slices, yielding a compact representation suitable for 2D convolutions by flattening the polarity and temporal dimensions.

**Problem Formulation.** Given two consecutive frames $F_a$ and $F_b$ captured at timestamps $T_a$ and $T_b$, our objective is to leverage the event stream over the interval $[T_a, T_b]$ to detect objects at the end timestamp $T_b$. Existing event-based object detection methods often use fixed time intervals $\Delta T$, limiting adaptability to dynamic environments (Perot et al., 2020). Additionally, integrating spatial information from RGB frames remains challenging, affecting performance in complex scenarios (De Tournemire et al., 2020). To address this, we synchronize event data with frames and explore varying training frequencies, leveraging the temporal richness of event cameras to improve detection accuracy.

## 3.2 FLEXFUSER: ADAPTIVE EVENT-FRAME FUSION MODULE

In dynamic environments, object detection systems must adapt to varying motion frequencies (Sun et al., 2022a). While event cameras excel at capturing rapid changes in pixel intensity, they often

lack the rich spatial and semantic information provided by frames. To address this limitation and fully leverage the complementary strengths of both modalities, we introduce **FlexFuser**, an adaptive fusion module designed to dynamically combine event data at different frequencies with frame data.

**Dynamic Event Aggregation.** Given a dataset $\mathcal{D}$, consisting of sequences of calibrated event camera data and frame data with a resolution of $H \times W$, along with corresponding bounding box annotations $y$ collected at frequency $a$, we begin by selecting a batch of frame data $\mathbf{F}$ paired with event data $\mathbf{E}^a$, both captured at frequency $a$. To aggregate event data from a higher frequency $b$ (where $b > a$), we divide the time interval $\Delta T^a$ corresponding to $\mathbf{E}^a$ into $b/a$ smaller sub-intervals. From each sub-interval, we obtain a high-frequency event set $\{\mathbf{E}_i^b\}_{i=0}^{b/a}$, as defined in Eq. 2. From this set, we randomly sample one event data point[1] $\mathbf{E}^b$. By doing so, the sampled high-frequency event data $\mathbf{E}^b$ is temporally aligned with the frame data $\mathbf{F}$ and the base frequency event data $\mathbf{E}^a$. This synchronization of event streams at different frequencies ensures consistent and reliable processing for subsequent stages.

**Feature Extraction.** Let $\phi_{\mathrm{E}}(\cdot)$ and $\phi_{\mathrm{F}}(\cdot)$ represent the event- and frame-based networks, respectively, where the former employs the RVT (Gehrig & Scaramuzza, 2023) for extracting features from event data, and the latter uses ResNet-50 (He et al., 2016) for feature extraction from frames. Both networks are structured into four stages, as shown in Fig. 2.

At each scale $i$, we extract the corresponding features $^{(i)}\mathbf{h}_{\mathrm{E}}^a, ^{(i)}\mathbf{h}_{\mathrm{E}}^b$ from the event data and $^{(i)}\mathbf{h}_{\mathrm{F}}$ from the frame data:

$$^{(i)}\mathbf{h}_{\mathrm{E}}^a = \phi_{\mathrm{E}}^{(i)}(\mathbf{E}^a), \quad ^{(i)}\mathbf{h}_{\mathrm{E}}^b = \phi_{\mathrm{E}}^{(i)}(\mathbf{E}^b), \quad ^{(i)}\mathbf{h}_{\mathrm{F}} = \phi_{\mathrm{F}}^{(i)}(\mathbf{F}), \quad (3)$$

where $^{(i)}\mathbf{h}_{\mathrm{E}}^a$ and $^{(i)}\mathbf{h}_{\mathrm{E}}^b \in \mathbb{R}^{B \times C_{\mathrm{E}} \times H_i \times W_i}$, $^{(i)}\mathbf{h}_{\mathrm{F}} \in \mathbb{R}^{B \times C_{\mathrm{F}} \times H_i \times W_i}$. Here, $i$ denotes the scale, $B$ is the batch size, and $C_{\mathrm{E}}$ and $C_{\mathrm{F}}$ are the dimensions of the feature maps extracted from the event and frame data, respectively.

**Event-Frame Adaptive Fuser.** To effectively fuse the event and frame data, we employ an adaptive fuser that is consistent across different event data frequencies. At each scale $i$, taking the low frequency event features $^{(i)}\mathbf{h}_{\mathrm{E}}^a$ as an example, we concatenate the feature maps from both the event and frame branches as follows:

$$^{(i)}\mathbf{h}_{\mathrm{shared}}^a = \left[ ^{(i)}\mathbf{h}_{\mathrm{E}}^a, \ ^{(i)}\mathbf{h}_{\mathrm{F}} \right] \in \mathbb{R}^{B \times (C_{\mathrm{E}}+C_{\mathrm{F}}) \times H_i \times W_i} \ . \quad (4)$$

Inspired by previous works (Zhou et al., 2023) and (Zhong et al., 2024), our goal is to dynamically fuse these two modalities in a flexible manner. The proposed FlexFuser module computes adaptive soft weights that regulate the contribution of each branch (event and frame) based on the current input conditions. As shown in Fig. 3, these adaptive soft weights are computed using a gating function, which incorporates learned noise to introduce perturbation for improved adaptability. The process is:

$$\left[ ^{(i)}\boldsymbol{\alpha}, \ ^{(i)}\boldsymbol{\beta} \right] = \mathrm{Softmax}\left( \left( ^{(i)}\mathbf{h}_{\mathrm{shared}}^a \cdot ^{(i)}\mathbf{W}_g \right) + ^{(i)}\sigma \cdot \epsilon \right), \quad (5)$$

where $^{(i)}\mathbf{W} \in \mathbb{R}^{(C_{\mathrm{E}}+C_{\mathrm{F}}) \times 2}$ is a trainable weight matrix, $^{(i)}\boldsymbol{\alpha}$ and $^{(i)}\boldsymbol{\beta}$ are the adaptive soft weights for the event and frame branches, respectively. Here, $^{(i)}\sigma$ is a learned standard deviation that controls the magnitude of the noise perturbation, and $\epsilon \sim \mathcal{N}(0,1)$ represents a Gaussian noise term.

The fused feature map at each scale $i$ is then obtained by applying the adaptive soft weights to the event and frame features:

$$^{(i)}\mathbf{h}_{\mathrm{fuse}}^a = ^{(i)}\boldsymbol{\alpha} \odot ^{(i)}\mathbf{h}_{\mathrm{E}}^a + ^{(i)}\boldsymbol{\beta} \odot ^{(i)}\mathbf{h}_{\mathrm{F}}, \quad (6)$$

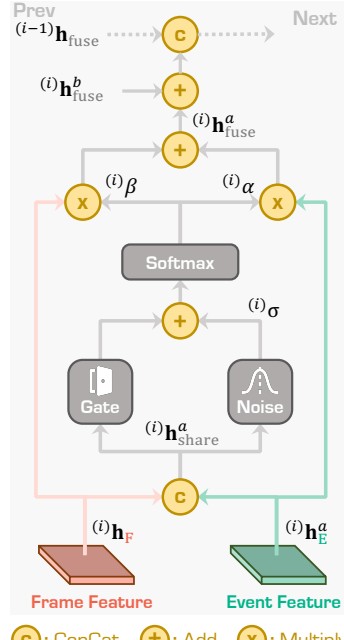

Figure 3: Illustration of the **Flex-Fuser** module. We show a general example of event and frame under frequency $a$ at stage $i$.

---

[1]For simplicity, we use $\mathbf{E}^b$ to represent a sample from the set of high-frequency event data $\{\mathbf{E}_i^b\}_{i=0}^{b/a}$, rather than explicitly referencing each individual sample from the event set. The same applies to other frequencies.

where $\odot$ denotes element-wise multiplication. This fusion process dynamically balances the contribution of each modality based on the input data, allowing for more robust and adaptive feature representation across varying conditions.

Then, at each scale $i$, the final feature map combines event data at different frequencies and the frame data is obtained by adding the fused features from the different frequencies. Specifically, we combine the fused feature maps as ${}^{(i)}\mathbf{h}_{\text{fuse}} = {}^{(i)}\mathbf{h}_{\text{fuse}}^a + {}^{(i)}\mathbf{h}_{\text{fuse}}^b$. After obtaining the fused feature maps across all scales, the multi-scale features are concatenated and fed into the detection head to produce the predicted bounding box $\hat{\mathbf{y}}$.

**Optimization & Regularization.** In addition to the standard detection loss $\mathcal{L}_{\text{det}}(\mathbf{y}, \hat{\mathbf{y}})$, such as the one used in YOLOX, we introduce a regularization term to ensure balanced utilization of both the event and RGB branches. This term penalizes large variations in the soft weights, encouraging a more uniform contribution from both modalities and preventing overfitting to a single branch:

$$\mathcal{L}_{\text{fuser}} = \mathcal{L}_{\text{det}}(\mathbf{y}, \hat{\mathbf{y}}) + \lambda \left( \frac{\text{Var}(\boldsymbol{\alpha})}{(\mathbb{E}[\boldsymbol{\alpha}])^2} + \frac{\text{Var}(\boldsymbol{\beta})}{(\mathbb{E}[\boldsymbol{\beta}])^2} \right), \tag{7}$$

where $\lambda$ is a weighting factor.

### 3.3 FAL: Frequency-Adaptive Learning Mechanism

FlexFuser aggregates information from different frequencies using labeled low-frequency data. To adaptively tune the model to handle diverse frequencies by leveraging both labeled low-frequency data and unlabeled high-frequency data, we design a flexible frequency-adaptive learning (**FAL**) mechanism. FAL incorporates multi-frequency information into the training process through iterative self-training. This approach enhances the model's ability to generalize across varying frequencies, making it more robust in different scenarios. The key steps of the FAL mechanism are as follows:

**Pre-Training with Low-Frequency Labels.** Rather than training solely at the same frequency as the data collection frequency $a$, we enhance the model's capability by training it at a higher frequency $b$. To efficiently leverage the available labels, we select only the final event from the high-frequency event set $\{\mathbf{E}_i^b\}_{i=0}^{b/a}$, which corresponds to the labeled timestamp. This approach allows the model to capture valuable high-frequency temporal information while still utilizing low-frequency labels, improving its temporal understanding and robustness. The training objective is to minimize the detection loss over the labeled data:

$$\mathcal{L}_{\text{GT}} = \sum_{(\mathbf{F}, \mathbf{E}_{b/a}^b, \mathbf{y}) \in \mathcal{D}} \mathcal{L}_{\text{det}}(\mathbf{y}, \hat{\mathbf{y}}). \tag{8}$$

**Label at Higher Frequencies.** For the unlabeled data in $\mathcal{D}$ captured at frequency $b$, the pre-trained model generates high-frequency labels $\hat{\mathbf{y}}$ by performing inference on the entire high-frequency event set $\{\mathbf{E}_i^b\}_{i=0}^{b/a}$. These generated labels $\hat{\mathbf{y}}$ serve as labels for guiding further training at higher frequencies, improving the model's ability to generalize across different temporal conditions.

**Enhanced Temporal Refinement.** To refine the high-frequency labels, we introduce a multi-step temporal refinement approach. First, we adopt bidirectional event augmentation by processing both forward and reversed event streams to detect objects with varying movements and orientations, thereby boosting recall. After generating the bidirectional labels, we apply Non-Maximum Suppression (NMS) to remove overlapping bounding boxes, followed by a low confidence threshold $\tau$ to retain potential objects and further improve recall. Next, leveraging a tracking-by-detection framework, we link detection boxes across frames using pairwise IoU matching with a threshold $\tau^{\text{IoU}}$. Short-lived tracks, with lengths below $\mathbf{L}^{\text{track}}$, are pruned to ensure temporal consistency. This approach ensures that the refined high-frequency labels $\tilde{\mathbf{y}}$ are accurate, temporally consistent, and reliable, ultimately improving detection quality in high-frequency data even in the absence of ground truth labels.

**Self-Training Iteration.** The model is iteratively trained using these refined high frequency labels $\tilde{\mathbf{y}}$ on high-frequency data where no ground truth labels are available. The total loss function combines the base training loss and the pseudo-label loss as:

$$\mathcal{L}_{\text{FAL}} = \mathcal{L}_{\text{GT}} + \beta \sum_{(\mathbf{F}, \{\mathbf{E}_i^b\}_{i=0}^{b/a-1}, \tilde{y}) \in \mathcal{D}} \mathcal{L}_{\text{det}}(\tilde{\mathbf{y}}, \hat{\mathbf{y}}), \tag{9}$$

Table 1: Comparative study of state-of-the-art event camera detectors on the validation set of *DSEC-Det* (Gehrig & Scaramuzza, 2024). Both event-only and event-frame fusion methods are compared. The **best** and 2nd best scores from each metric are highlighted in **bold** and underlined, respectively.

| Modality | Method | Venue | Reference | mAP | AP$_{50}$ | AP$_{75}$ | AP$_S$ | AP$_M$ | AP$_L$ |
|---|---|---|---|---|---|---|---|---|---|
| E | RVT | CVPR'23 | (Gehrig & Scaramuzza, 2023) | 38.4% | 58.7% | 41.3% | 29.5% | 50.3% | 81.7% |
| | SAST | CVPR'24 | (Peng et al., 2024) | 38.1% | 60.1% | 40.0% | 29.8% | 48.9% | 79.7% |
| | SSM | CVPR'24 | (Zubić et al., 2024) | 38.0% | 55.2% | 40.6% | 28.8% | 52.2% | 77.8% |
| | LEOD | CVPR'24 | (Wu et al., 2024) | 41.1% | 65.2% | 43.6% | 35.1% | 47.3% | 73.3% |
| E + F | DAGr-18 | Nature'24 | (Gehrig & Scaramuzza, 2024) | 37.6% | - | - | - | - | - |
| | DAGr-34 | Nature'24 | (Gehrig & Scaramuzza, 2024) | 39.0% | - | - | - | - | - |
| | DAGr-50 | Nature'24 | (Gehrig & Scaramuzza, 2024) | 41.9% | 66.0% | 44.3% | 36.3% | 56.2% | 77.8% |
| | FlexEvent | **Ours** | - | 57.4% | **78.2%** | **66.6%** | **51.7%** | **64.9%** | **83.7%** |

where $\beta$ balances the contribution of the high-frequency label loss. The complete FlexEvent framework combines FlexFuser and FAL, allowing the model to dynamically fuse event and frame data while adapting to varying frequencies. As we will verify in the next sections, this combination provides a robust detection framework capable of maintaining high accuracy in dynamic environments.

# 4 EXPERIMENTS

## 4.1 EXPERIMENTAL SETTINGS

**Datasets.** We conduct experiments based on three large-scale event camera datasets: [1]*DSEC-Det* (Gehrig & Scaramuzza, 2024), [2]*DSEC-Detection* (Tomy et al., 2022), and [3]*DSEC-MOD* (Zhou et al., 2023). These datasets comprise $78,344$ frames across 60 sequences, $52,727$ frames over 41 sequences, and $13,314$ frames within 16 sequences, respectively, making them suitable for evaluating event-based object detection methods. For more details, please refer to the Appendix.

**Implementation Details.** We trained our model using the YOLOX framework (Zheng et al., 2021), optimizing with a combination of IoU loss, classification loss, and regression loss, averaged across both batch and sequence length for stable training. We also introduce the extra MoE loss for balancing the utilization among the experts. The model was trained for $100,000$ iterations with a batch size of 8 and a sequence length of 11, using a learning rate of $1 \times 10^{-4}$. All experiments were conducted on two NVIDIA RTX A5000 GPUs with 24GB memory, with the entire training process completed in approximately one day. Due to space limits, more details are placed in the Appendix.

**Evaluation Metrics.** We evaluate object detectors using the mean Average Precision (mAP) as the primary metric, along with AP$_{50}$, AP$_{75}$, AP$_S$, AP$_M$, and AP$_L$ from the COCO evaluation protocol (Lin et al., 2014). These metrics provide a comprehensive assessment of detection performance across different IoU thresholds and object sizes. Kindly refer to the Appendix for more details.

## 4.2 COMPARISONS TO STATE-OF-THE-ART DETECTORS

**Compare to Event-Only Models.** We compared FlexEvent with state-of-the-art event-only detectors, including RVT (Gehrig & Scaramuzza, 2023), SSM (Zubić et al., 2024), SAST (Peng et al., 2024), and LEOD (Wu et al., 2024), as shown in Tab. 1. We significantly outperform these methods across all metrics, with the performance gap becoming even more pronounced at higher frequencies. Event-only methods struggle to maintain detection accuracy in these scenarios due to their inability to fully capture object semantics. In contrast, we overcome these limitations through the FlexFuser module, which integrates RGB data to compensate for the lack of semantic richness in the event stream. By fusing both event and frame data, we excel in complex, dynamic environments, achieving superior detection accuracy where event-only methods fall short.

**Compare to Multimodal Models.** We compare FlexEvent with multimodal event-camera object detection methods such as DAGr (Gehrig & Scaramuzza, 2024) and SPNet (Zhou et al., 2021), which fuse event data with other sensor inputs like RGB frames or depth to improve detection accuracy. While these methods enhance performance over event-only approaches, they struggle with adapting to varying operational frequencies and often exhibit inadequate feature fusion in dynamic environments. Our approach addresses these limitations by dynamically balancing the contributions of event and

Table 2: Comparative study of state-of-the-art event camera detectors on the test set of *DSEC-Detection* (Tomy et al., 2022). Both event-only and event-frame fusion methods are compared. The reported results are the **mAP** scores of [1]Car, [2]Pedestrian (Ped), and [3]Large-Vehicle (L-Veh) classes. The **best** and 2nd best scores from each metric are highlighted in **bold** and underlined, respectively.

| Modality | Method | Venue | Reference | Type | Car | Ped | L-Veh | Average |
|---|---|---|---|---|---|---|---|---|
| E | CAFR | ECCV'24 | (Cao et al., 2024) | Event | - | - | - | 12.0% |
| E + F | SENet | CVPR'18 | (Hu et al., 2018) | Attention | 38.4% | 14.9% | 26.0% | 26.2% |
| | CBAM | ECCV'18 | (Woo et al., 2018) | | 37.7% | 13.5% | 27.0% | 26.1% |
| | ECA-Net | CVPR'20 | (Wang et al., 2020) | | 36.7% | 12.8% | 27.5% | 25.7% |
| | SAGate | ECCV'20 | (Chen et al., 2020) | RGB + Depth | 32.5% | 10.4% | 16.0% | 19.6% |
| | DCF | CVPR'21 | (Ji et al., 2021) | | 36.3% | 12.7% | 28.0% | 25.7% |
| | SPNet | ICCV'21 | (Zhou et al., 2021) | | 39.2% | 17.8% | 26.2% | 27.7% |
| | RAMNet | RA-L'21 | (Gehrig et al., 2021a) | RGB + Event | 24.4% | 10.8% | 17.6% | 17.6% |
| | FAGC | Sensors'21 | (Cao et al., 2021) | | 39.8% | 14.4% | 33.6% | 29.3% |
| | FPN-Fusion | ICRA'22 | (Tomy et al., 2022) | | 37.5% | 10.9% | 24.9% | 24.4% |
| | EFNet | ECCV'22 | (Sun et al., 2022a) | | 41.1% | 15.8% | 32.6% | 30.0% |
| | DRFuser | EAAI'23 | (Munir et al., 2023) | | 38.6% | 15.1% | 30.6% | 28.1% |
| | CMX | TITS'23 | (Zhang et al., 2023) | | 41.6% | 16.4% | 29.4% | 29.1% |
| | RENet | ICRA'23 | (Zhou et al., 2023) | | 40.5% | 17.2% | 30.6% | 29.4% |
| | CAFR | ECCV'24 | (Cao et al., 2024) | | 49.9% | 25.8% | 38.2% | 38.0% |
| | FlexEvent | **Ours** | - | | **59.3%** | **37.4%** | **45.5%** | **47.4%** |

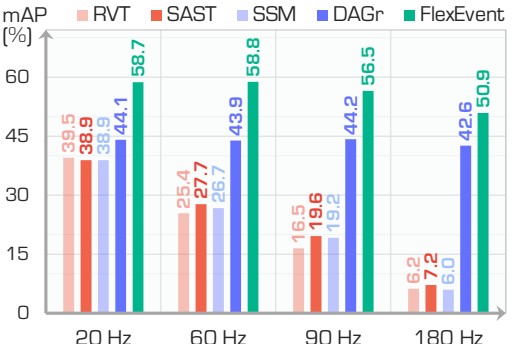

Figure 4: Comparisons of event-based detectors under different event frequencies on *DSEC-Det*.

Table 3: Comparisons of fusion-based event detectors on the test set of *DSEC-MOD* (Zhou et al., 2023). The **best** and 2nd best scores from each metric are highlighted in **bold** and underlined.

| Method | Venue | Reference | mAP |
|---|---|---|---|
| SENet | CVPR'18 | (Hu et al., 2018) | 29.28% |
| CBAM | ECCV'18 | (Woo et al., 2018) | 36.22% |
| ECA-Net | CVPR'20 | (Wang et al., 2020) | 34.49% |
| SAGate | ECCV'20 | (Chen et al., 2020) | 33.62% |
| DCF | CVPR'21 | (Ji et al., 2021) | 32.20% |
| SPNet | ICCV'21 | (Zhou et al., 2021) | 32.70% |
| FPN-Fusion | ICRA'22 | (Tomy et al., 2022) | 32.28% |
| EFNet | ECCV'22 | (Sun et al., 2022a) | 35.33% |
| RENet | ICRA'23 | (Zhou et al., 2023) | 38.38% |
| FlexEvent | **Ours** | - | **48.64%** |

frame data. As a result, we achieve superior performance, such as a $48.64\%$ mAP on *DSEC-MOD* in Tab. 3, outperforming RENet ($38.38\%$) and EFNet ($35.33\%$). This flexible combination of event and frame data, along with its ability to generalize across different temporal resolutions, enables us to excel in high-frequency detection scenarios, surpassing state-of-the-art methods.

**Comparisons Across Different Categories.** We evaluate the performance of FlexEvent across various object categories, including cars, pedestrians, and large vehicles. As shown in Tab. 2, we consistently outperform other methods, achieving $59.3\%$ mAP for cars, compared to $49.9\%$ for CAFR (Cao et al., 2024). This highlights its effectiveness in detecting larger, fast-moving objects, while also surpassing CAFR on pedestrian and large-vehicle categories. Existing methods struggle with smaller, slower-moving objects, especially at high speeds. Event-based detectors like SSM miss stationary objects due to a lack of pixel intensity changes, and fusion methods over-rely on frame data, which lacks temporal resolution. Our approach addresses these issues with adaptive fusion and temporal refinement, ensuring accurate detection across different object types and motion dynamics. This versatility reinforces its superiority over state-of-the-art methods.

**Generalization on High-Frequency Data.** A key contribution of FlexEvent is its ability to generalize across various operational frequencies, particularly in high-frequency scenarios. We evaluate this by testing detection performance at different temporal offsets, $\frac{i}{n}\Delta T$, where $n = 10$, $i = 0, ..., 10$, and $\Delta T = 50$ ms. Ground truth labels are generated by linearly interpolating object positions between frames for consistent evaluation. In this setting, event-based methods are tested across multiple time durations, while event-frame fusion methods process one RGB frame followed by event data of varying time durations. The comparison result is shown in Fig. 4. Most existing methods, such as

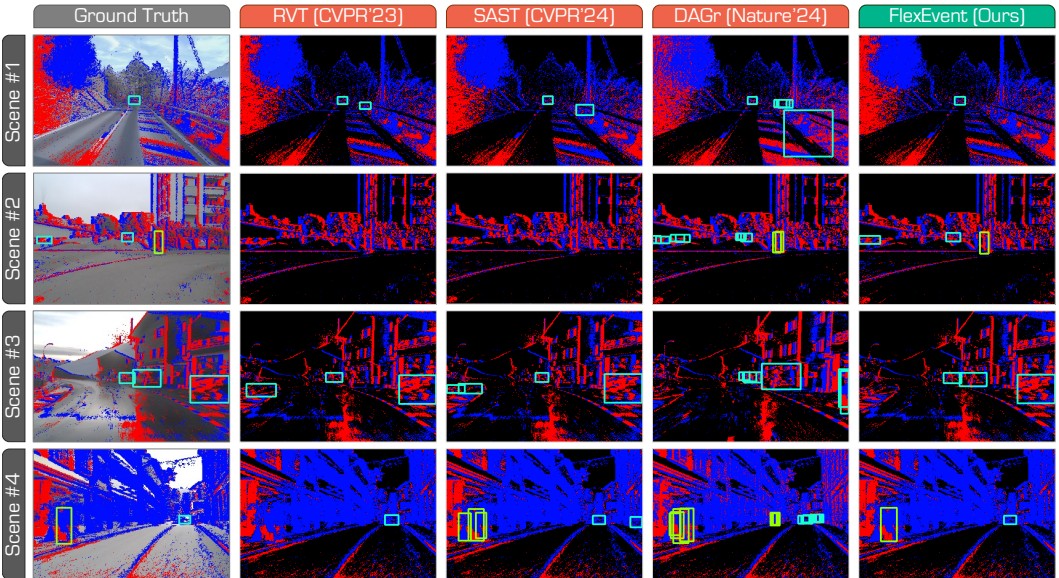

Figure 5: Qualitative results of state-of-the-art event camera detectors. We compare FlexEvent with RVT (Gehrig & Scaramuzza, 2023), SAST (Peng et al., 2024), and DAGr (Gehrig & Scaramuzza, 2024) on the validation set of *DSEC-Det*. Best viewed in colors. See the Appendix for more examples.

Table 4: Ablation study of components in FlexEvent. **EFF** denotes the adaptive event-frame fusion module (*cf.* Sec.3.2). **FAL** denotes the frequency-adaptive learning module (*cf.* Sec. 3.3). The reported are the **mAP** scores on the test set of *DSEC-Det* (Gehrig & Scaramuzza, 2024). The symbol ⬧ denotes the use of interpolated ground truth labels at high frequencies in **FAL**.

| Modality | FAL | EFF | 20.0 | 27.5 | 30.0 | 36.0 | 45.0 | 60.0 | 90.0 | 180 | Average |
|---|---|---|---|---|---|---|---|---|---|---|---|
| | | | \multicolumn{8}{c}{**Frequency (Hz)**} | |
| E | ✗ | ✗ | 53.2% | 54.0% | 53.5% | 52.0% | 49.4% | 45.9% | 38.8% | 22.9% | 46.2% |
| | ✓ | ✗ | 54.6% | 54.9% | 54.9% | 54.3% | 53.3% | 50.7% | 44.6% | 30.4% | 49.7% |
| E + F | ⬧ | ✓ | 54.9% | 57.3% | 57.7% | 57.8% | 57.2% | 56.1% | 53.7% | 48.3% | 55.4% |
| | ✗ | ✓ | **58.0%** | 59.6% | **60.0%** | 59.6% | 59.0% | 57.6% | 54.8% | 49.2% | 57.2% |
| | ✓ | ✓ | 57.4% | **60.0%** | **60.0%** | **60.1%** | **59.5%** | **58.8%** | **56.5%** | **50.9%** | **57.9%** |

RVT and SAST, struggle at higher frequencies due to fixed temporal intervals and limited ability to capture fast scene changes. In contrast, our approach achieves 56.5% mAP at 90 Hz and 50.9% at 180 Hz. This improvement demonstrate that our method excels in dynamic, rapidly changing environments where accurate detection is critical for safety and reliability.

**Qualitative Assessment.** We provide qualitative comparisons between FlexEvent and other state-of-the-art methods under different event operation frequencies, as shown in Fig. 5 and Fig. 6, with visual results from *DSEC-Det* highlighting our superior detection capabilities. Unlike RVT and DAGr, which miss critical object details, our model consistently detects objects with high accuracy, even in challenging cases involving fast-moving vehicles and occluded pedestrians. For instance, in Scene 2 of Fig. 5, RVT fails to detect a pedestrian due to insufficient event data, while our approach successfully identifies the pedestrian by leveraging both frames and high-frequency event data. Similarly, in Scene 4, DAGr struggles with the rapid motion of a large vehicle, leading to inaccurate predictions, whereas our approach ensures precise object localization. These qualitative findings confirm that our model excels not only in quantitative metrics but also in real-world performance.

## 4.3 ABLATION STUDIES

**Component Analysis.** We conduct ablation studies by selectively removing key modules: the FlexFuser and FAL mechanisms. As shown in Tab. 4, removing the FAL mechanism causes a significant performance drop, particularly in high-frequency scenarios, underscoring its role in adapting to varying frequencies and generating frequency-adjusted labels. Similarly, omitting the FlexFuser module leads to a marked decrease in mAP, highlighting the importance of adaptive event-

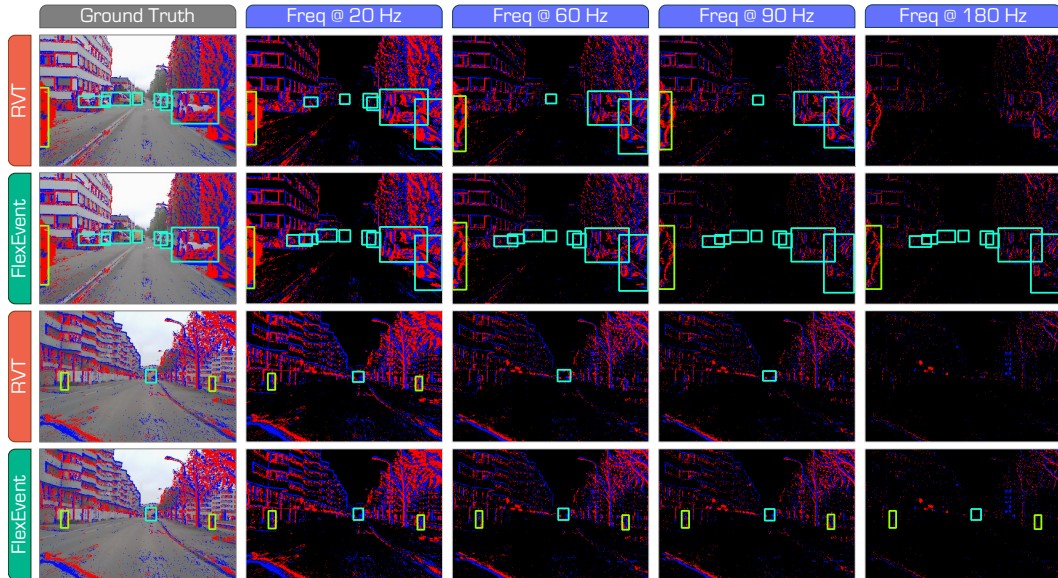

Figure 6: Qualitative comparisons of FlexEvent and RVT (Gehrig & Scaramuzza, 2023) under different event operation frequencies. Our approach demonstrates a strong robustness under both low- and high-frequency scenarios. Best viewed in colors. See the Appendix for more examples.

Table 5: Ablation study of hyperparameter configurations in the FlexEvent frameworks. $\tau^{car}$, $\tau^{ped}$ denotes the confidence threshold for car and pedestrian, respectively. $\tau^{iou}$ denotes the IoU threshold when filter by tracking, $\mathbf{L}^{track}$ denotes the minimum track length. The reported results are the **mAP** scores on the validation set of *DSEC-Det* (Gehrig & Scaramuzza, 2024).

| $\tau^{car}$ | $\tau^{ped}$ | $\mathbf{L}^{track}$ | $\tau^{iou}$ | Frequency (Hz) | | | | | | | | Average |
| | | | | 20.0 | 27.5 | 30.0 | 36.0 | 45.0 | 60.0 | 90.0 | 180 | |
|---|---|---|---|---|---|---|---|---|---|---|---|---|
| 0.6 | 0.3 | 10 | 0.8 | 56.5% | 55.9% | 56.7% | 57.2% | 57.1% | 56.7% | 54.5% | 49.2% | 55.5% |
| 0.6 | 0.3 | 10 | 0.6 | 56.7% | 57.2% | 57.7% | 57.9% | 57.7% | 57.0% | 54.3% | 47.0% | 55.7% |
| 0.6 | 0.3 | 8 | 0.6 | 56.3% | 58.5% | 58.8% | 59.1% | 58.8% | 58.4% | 56.2% | **51.2%** | 57.2% |
| 0.6 | 0.3 | 6 | 0.6 | 57.3% | 59.4% | 59.7% | 59.9% | 59.3% | 58.5% | 55.7% | 48.8% | 57.3% |
| 0.6 | 0.6 | 6 | 0.6 | 57.4% | 60.0% | 60.0% | 60.1% | 59.5% | 58.8% | 56.5% | 50.9% | **57.9%** |
| 0.8 | 0.8 | 6 | 0.6 | 56.6% | 58.7% | 59.1% | 58.9% | 58.4% | 57.4% | 55.6% | 50.2% | 56.9% |

frame fusion for accurate detection across different operational frequencies. We also test training with interpolation labels for high-frequency testing, but this approach reduces recall by missing objects that suddenly appear or disappear, making it less effective than FAL.

**Hyperparameter Tuning.** We tune the hyperparameters of the FAL mechanism, focusing on key settings like the confidence threshold ($\tau$), IoU threshold, and track length for temporal refinement. As shown in Tab. 5, lowering the confidence threshold improves recall but reduces precision, as the model becomes more lenient in detecting objects. Applying overly strict conditions, such as a higher confidence threshold or IoU threshold, lowers recall by filtering out valid detections. The optimal configuration is achieved with $\tau = 0.6$ and a track length of 6, balancing precision and recall for both low- and high-frequency conditions. These moderate settings ensure that FlexEvent maintains robust performance and stable detection accuracy across diverse environments.

## 5 CONCLUSION

This paper introduces FlexEvent, an event camera object detection framework designed to operate across arbitrary frequencies. By combining FlexFuser for adaptive event-frame fusion and FAL for frequency-adaptive learning, we combine event data's rich temporal information with the semantic detail of RGB frames to overcome the limitations of existing methods and offer a flexible solution for dynamic environments. Extensive experiments on large-scale datasets show that our approach significantly outperforms state-of-the-art methods, particularly in high-frequency scenarios, demonstrating its robustness and adaptability for real-world applications like autonomous driving and robotics.

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

# APPENDIX

In this appendix, we supplement the following materials to support the findings and conclusions drawn in the main body of this paper.

## A  ADDITIONAL IMPLEMENTATION DETAILS

In this section, to facilitate future reproductions, we elaborate on the necessary details in terms of the datasets, evaluation metrics, and implementation details adopted in our experiments.

### A.1  DATASETS

In this work, we develop and validate our proposed approach on the large-scale DSEC dataset (Gehrig et al., 2021b). DSEC serves as a high-resolution, large-scale multimodal dataset designed to capture real-world driving scenarios under various conditions. It combines data from stereo Prophesee Gen3 event cameras with a resolution of $640 \times 480$ pixels and FLIR Blackfly S RGB cameras operating at 20 FPS, enabling high-fidelity capture of dynamic scenes. To align the RGB frames with the event camera data, an infinite-depth alignment process is employed, which involves undistorting, rotating, and re-distorting the RGB images. This alignment ensures that the event data and RGB frames are temporally and spatially synchronized.

Table 6: Summary of key statistics from the event camera object detection datasets used in this work.

| Dataset | Reference | Classes | Frames | Sequences | Class Names |
|---|---|---|---|---|---|
| DSEC-MOD | (Zhou et al., 2023) | 1 | 13,314 | 16 | Car |
| DSEC-Detection | (Tomy et al., 2022) | 3 | 52,727 | 41 | Car
Pedestrian
Large-Vehicle |
| DSEC-Det | (Gehrig & Scaramuzza, 2024) | 8 | 78,344 | 60 | Car
Pedestrian
Bus
Bicycle
Truck
Motorcycle
Rider
Train |

In our experiments, we utilize **three** comprehensive versions of DSEC tailored for object detection: *DSEC-Det* (Gehrig & Scaramuzza, 2024), *DSEC-Detection* (Tomy et al., 2022), and *DSEC-MOD* (Zhou et al., 2023). A summary of the key statistics of these datasets is listed in Tab. 6.

- **DSEC-Det** (Gehrig & Scaramuzza, 2024): This version was developed by the original DSEC team and includes annotations generated using the QDTrack multi-object tracker (Fischer et al., 2023; Pang et al., 2021). The annotation process involved tracking multiple objects across frames, followed by manual refinement to ensure high-quality and accurate detection labels. The dataset introduces additional sequences specifically designed to capture complex, dynamic urban environments, featuring crowded pedestrian areas, moving vehicles, and diverse lighting conditions. These challenging scenarios provide a rich testing ground for evaluating object detection algorithms in real-world driving settings. In total, DSEC-Det features 60 sequences comprising 78,344 frames, making it the most extensive dataset used in this study. It captures diverse, complex urban scenes with dynamic environments, such as crowded pedestrian areas and moving vehicles. Covering eight object categories relevant to autonomous driving – Car, Pedestrian, Bus, Bicycle, Truck, Motorcycle, Rider, and Train – this dataset provides a robust foundation for training and evaluating object detection models in diverse driving scenarios. In our experiment on DSEC-Det, to be consistent with the experiment setting of previous work DAGr (Gehrig & Scaramuzza, 2024), we report results on two categories: Car and Pedestrian.

- **DSEC-Detection** (Tomy et al., 2022): The dataset comprises 41 sequences with a total of 52,727 frames. Focusing on three fundamental object categories – Car, Pedestrian, and Large-Vehicle – this version emphasizes high-precision annotations for these critical classes in autonomous driving. The initial annotations were generated using the YOLOv5 model (Jocher, 2020) on RGB frames, known for its robust performance in real-time object detection. These annotations were then transferred to the corresponding event frames through homographic transformation, ensuring spatial alignment between the two modalities. A subsequent manual refinement process was conducted to correct any discrepancies and improve annotation quality, resulting in a dataset that provides accurate and reliable labels for event-based object detection.

- **DSEC-MOD** (Zhou et al., 2023): As one of the most recent and comprehensive versions, DSEC-MOD extends the object detection capabilities to multi-object detection across diverse urban environments. It includes 16 sequences containing 13,314 frames and is specifically focused on the Car category, making it highly suitable for complex detection tasks in varied urban settings, such as intersections, highways, and residential areas. The dataset features high-frequency and dense annotations, providing a valuable resource for evaluating event-based object detectors' performance under challenging real-world conditions.

These three versions of the DSEC dataset together offer a comprehensive platform for benchmarking and evaluating event-based object detection methods, capturing a wide spectrum of scenarios, object categories, and environmental conditions. Among them, DSEC-Det is the most recent, largest, and

most comprehensive one, annotated and released by the original DSEC authors. Thus, we prioritized it as the primary benchmark for reporting results, ensuring relevance and reliability. DSEC-Detection and DSEC-MOD are datasets used by two recent event-frame fusion methods CAFR (Cao et al., 2024) and RENet (Zhou et al., 2023), so we also report results on these two datastes to validate our method's effectiveness.

## A.2 BASELINES

To evaluate the effectiveness of our method, we compare it against both event-only and event-frame fusion state-of-the-art methods.

**Event-Only Methods.** We include state-of-the-art event-only object detectors, namely RVT (Gehrig & Scaramuzza, 2023), SAST (Peng et al., 2024), LEOD (Wu et al., 2024), and SSM (Zubić et al., 2024), which are originally trained on event-only datasets like Gen1 (De Tournemire et al., 2020) and 1Mpx (Perot et al., 2020). To ensure a fair comparison, we retrain these methods on the DSEC-Det dataset following their respective training protocols.

**Event-Frame Fusion Methods.** For event-frame fusion methods on DSEC-Det, we include DAGr, as it has been evaluated on this dataset. We report the scores of DAGr (Gehrig & Scaramuzza, 2024) from the original paper to ensure consistency and fairness. For the DSEC-Detection and DSEC-MOD datasets, we train our model following the standard training and evaluation settings. We compare our method against state-of-the-art methods CAFR (Cao et al., 2024) and RENet (Zhou et al., 2023), as reported in their respective papers. For other methods evaluated on DSEC-Detection and DSEC-MOD, we reference the results reported in the CAFR and RENet papers, respectively. Since DAGr's training code is not publicly available, we are unable to reproduce its results on DSEC-Detection and DSEC-MOD.

These comparisons ensure a fair and comprehensive evaluation while adhering to resource and code availability constraints.

## A.3 EVALUATION METRICS

In this work, we adopt the mean Average Precision (**mAP**) as the primary metric to evaluate the performance of our object detection models, consistent with standard practices in the field. The mAP metric provides a comprehensive measure of detection accuracy across multiple categories and intersection-over-union (IoU) thresholds.

Mathematically, the **Average Precision (AP)** for a single class is calculated as:

$$\text{AP} = \int_0^1 p(r) \, dr \tag{10}$$

where $p(r)$ represents the precision at a given recall level $r$. The mean Average Precision (mAP) is then computed as the mean of the AP values across all object categories and a range of IoU thresholds (typically from 0.5 to 0.95 with a step size of 0.05). This provides an overall measure of model performance across different levels of localization precision.

In addition to mAP, we also report the following metrics from the COCO evaluation protocol (Lin et al., 2014):

- **AP$_{50}$**: The average precision when evaluated at a fixed IoU threshold of 0.50, indicating how well the model performs with relatively lenient localization criteria.

- **AP$_{75}$**: The average precision at a fixed IoU threshold of 0.75, representing performance under stricter localization requirements.

- **AP$_\text{S}$, AP$_\text{M}$, and AP$_\text{L}$**: These metrics represent the average precision for small ($S$), medium ($M$), and large ($L$) objects, respectively. Object sizes are defined based on their pixel area, with **AP$_\text{S}$** typically representing objects with areas less than $32 \times 32$ pixels, **AP$_\text{M}$** representing areas between $32 \times 32$ and $96 \times 96$ pixels, and **AP$_\text{L}$** for objects larger than $96 \times 96$ pixels.

By reporting these metrics, we obtain a more nuanced understanding of the model's detection capabilities across varying object sizes and localization precision levels, ensuring a comprehensive evaluation of detection performance.

### A.4 TRAINING & INFERENCE DETAILS

We train our models using mixed precision to optimize both memory efficiency and training speed. The training process spans 100,000 iterations, utilizing the ADAM optimizer (Kingma, 2014) with a OneCycle learning rate schedule (Smith & Topin, 2019), which gradually decays from a peak learning rate to enhance convergence.

Consistent with (Gehrig & Scaramuzza, 2023), we employed a mixed batching strategy to balance computational efficiency and memory usage. Specifically:

- Standard Backpropagation Through Time (BPTT): Applied to half of the training samples, allowing for full sequence training.
- Truncated BPTT (TBPTT): Used for the other half, reducing memory usage by splitting sequences into smaller segments.

For data augmentation, we applied random horizontal flipping and zoom transformations (both zoom-in and zoom-out) to enhance the diversity of training samples.

Our training process utilized the YOLOX framework (Zheng et al., 2021), a versatile object detection framework known for its efficient and high-performing architecture. We employed a multi-component loss function to optimize our model effectively:

- Intersection over Union (IoU) Loss: This loss component measures the overlap between the predicted bounding boxes and the ground-truth boxes, ensuring that the predicted regions closely match the actual object locations.
- Classification Loss: This component evaluates the accuracy of class predictions for each detected object, ensuring that the model correctly identifies the category of each detected instance.
- Regression Loss: This loss assesses the precision of the predicted bounding box coordinates, helping the model refine the location and size of bounding boxes to align closely with the ground-truth annotations.

To ensure stable training, these loss components were averaged across both the batch and sequence length at each optimization step. This averaging process helped to reduce variance during training and facilitated smoother convergence of the model parameters.

**Training Configuration.** The training was conducted with a batch size of 8, which provided an optimal balance between efficient GPU utilization and memory requirements. Each training sample contained a sequence length of 11 frames, allowing the model to learn temporal dependencies effectively. The learning rate was set to $1 \times 10^{-4}$, following a OneCycle learning rate schedule that allowed for efficient exploration of the learning space and helped in achieving faster convergence.

**Hardware & Training Time.** All training experiments were carried out on two NVIDIA RTX A5000 GPUs, each with 24GB of memory, providing the computational resources necessary for handling the high-resolution event data and RGB frames. The complete training process, including all iterations and model optimization, took approximately one day, demonstrating the efficiency of our implementation in terms of both training speed and resource utilization.

## B ADDITIONAL QUANTITATIVE RESULTS

### B.1 COMPARISON OF EFFICIENCY

We present a comparative analysis of inference times and parameter counts for the evaluated methods in Tab. 7. All experiments were conducted on an NVIDIA RTX A5000 24GB GPU paired with an AMD EPYC 9354P 32-Core Processor operating at 3.8 GHz. The results demonstrate that, despite

Table 7: Comparative efficiency analysis of state-of-the-art event camera detectors on the validation set of *DSEC-Det* (Gehrig & Scaramuzza, 2024), comparing both event-only and event-frame fusion methods. This table reports **inference times** at various frequencies, measured in **milliseconds (ms)**.

| Modality | Method | Param (M) | Frequency (Hz) | | | | | | | | |
|----------|--------|-----------|------|------|------|------|------|------|------|------|------|
| | | | 20.0 | 27.5 | 30 | 36 | 45 | 60 | 90 | 180 | 200 |
| E | RVT | 18.5 | 9.20 | 8.67 | 8.35 | 7.93 | 7.61 | 7.51 | 7.19 | 6.77 | 6.34 |
| | SAST | 18.9 | 14.06 | 13.11 | 12.68 | 12.37 | 11.95 | 11.63 | 11.52 | 11.10 | 10.36 |
| | SSM | 18.2 | 8.79 | 8.26 | 8.08 | 7.71 | 7.55 | 7.30 | 6.90 | 6.54 | 6.12 |
| E + F | DAGr-50 | 34.6 | 73.35 | 65.73 | 60.02 | 55.11 | 51.00 | 48.00 | 45.29 | 43.89 | 37.58 |
| | FlexEvent | 45.4 | 14.27 | 13.53 | 13.32 | 13.00 | 12.79 | 12.58 | 12.47 | 12.37 | 12.12 |

Table 8: Comparison of FlexEvent with data augmentation methods EventDrop (Gu et al., 2021) and Shadow Mosaic (Peng et al., 2023). **FAL** represents the frequency-adaptive learning module (*cf.* Sec. 3.3), **Drop** refers to the EventDrop augmentation technique (Gu et al., 2021), and **Mosaic** corresponds to the Shadow Mosaic method (Peng et al., 2023). The table reports **mAP** scores evaluated on the test set of *DSEC-Det* (Gehrig & Scaramuzza, 2024).

| Drop | Mosaic | FAL | Frequency (Hz) | | | | | | | | Average |
|------|--------|-----|------|------|------|------|------|------|------|------|---------|
| | | | 20.0 | 27.5 | 30.0 | 36.0 | 45.0 | 60.0 | 90.0 | 180 | |
| ✗ | ✗ | ✗ | 53.2% | 54.0% | 53.5% | 52.0% | 49.4% | 45.9% | 38.8% | 22.9% | 46.2% |
| ✓ | ✗ | ✗ | 53.6% | 54.4% | 53.8% | 52.7% | 50.2% | 47.2% | 40.2% | 24.5% | 47.1% |
| ✗ | ✓ | ✗ | 53.7% | 54.4% | 54.0% | 53.9% | 51.4% | 48.6% | 41.8% | 27.8% | 48.2% |
| ✗ | ✗ | ✓ | **54.6%** | **54.9%** | **54.9%** | **54.3%** | **53.3%** | **50.7%** | **44.6%** | **30.4%** | **49.7%** |

having a higher parameter count, FlexEvent achieves inference times comparable to the event-only method SAST and significantly outperforms the event-frame fusion method DAGr in terms of speed. Moreover, FlexEvent consistently outperforms all other methods across all tested frequencies. These results underscore the efficiency and rapid performance of FlexEvent, highlighting its suitability for real-time applications.

## B.2 COMPARISON WITH DATA AUGMENTATION METHODS

We include the comparison of FlexEvent with data augmentation methods EventDrop (Gu et al., 2021) and Shadow Mosaic (Peng et al., 2023) with only Event modality in Tab. 8. EventDrop and Shadow Mosaic demonstrate good performance enhancement, credited to the strong generalization ability brought by the spatial and temporal manipulations of the event data. However, FAL significantly outperforms other methods by leveraging high-frequency event data, especially in high-frequency scenarios. The iterative refinement through self-training in our method ensures that the model remains robust across different motion dynamics and frequency settings.

## B.3 COMPLETE RESULTS OF ABLATION STUDY

We include the complete results of the ablation study in Tab. 9.

## B.4 COMPLETE RESULTS OF HYPERPARAMETER SEARCHING

We include the complete results of the hyperparameter searching in Tab. 10.

## C ADDITIONAL QUALITATIVE RESULTS

### C.1 VISUAL COMPARISONS OF EVENT CAMERA DETECTORS

We include additional qualitative assessments in Fig. 7, Fig. 8, and Fig. 9.

### C.2 VISUAL COMPARISONS UNDER DIFFERENT FREQUENCIES

We include additional qualitative assessments in Fig. 10, Fig. 11, and Fig. 12.

Table 9: The complete results of the ablation study (*cf.* Tab. 4) of different components in the FlexEvent framework. **EFF** denotes the adaptive event-frame fusion module (*cf.* Sec.3.2). **FAL** denotes the frequency-adaptive learning module (*cf.* Sec. 3.3). The reported results are the **mAP**, $AP_{50}$, $AP_{75}$, $AP_S$, $AP_M$, and $AP_L$ scores on the validation set of *DSEC-Det* (Gehrig & Scaramuzza, 2024). The symbol ◆ denotes the use of interpolated ground truth labels at high frequencies in **FAL**. The **best** and 2nd best scores of each metric are highlighted in **bold** and underline, respectively.

| Modality | FAL | EFF | Frequency (Hz) | mAP | $AP_{50}$ | $AP_{75}$ | $AP_S$ | $AP_M$ | $AP_L$ |
|---|---|---|---|---|---|---|---|---|---|
| Event | ✗ | ✗ | 20.0 | 53.2% | **77.2%** | 58.1% | **46.4%** | 64.4% | 83.0% |
| | ✗ | ✗ | 27.5 | **54.0%** | 76.8% | **59.3%** | **46.4%** | 66.6% | **85.2%** |
| | ✗ | ✗ | 30.0 | 53.5% | 75.5% | **59.3%** | 45.6% | **66.8%** | 85.0% |
| | ✗ | ✗ | 36.0 | 52.0% | 73.3% | 58.1% | 44.0% | 65.5% | 84.9% |
| | ✗ | ✗ | 45.0 | 49.4% | 69.5% | 55.4% | 40.7% | 64.1% | 84.3% |
| | ✗ | ✗ | 60.0 | 45.9% | 64.2% | 51.8% | 36.5% | 62.3% | 82.7% |
| | ✗ | ✗ | 90.0 | 38.8% | 55.4% | 43.9% | 28.5% | 55.3% | 79.9% |
| | ✗ | ✗ | 180.0 | 22.9% | 36.1% | 23.9% | 14.1% | 34.5% | 60.1% |
| Event | ✓ | ✗ | 20.0 | 54.6% | 79.1% | **61.8%** | 47.4% | 64.4% | 81.4% |
| | ✓ | ✗ | 27.5 | **54.9%** | 78.8% | 61.4% | **47.6%** | 66.1% | 83.2% |
| | ✓ | ✗ | 30.0 | **54.9%** | 78.5% | 61.3% | 47.4% | **66.9%** | 83.3% |
| | ✓ | ✗ | 36.0 | 54.3% | 77.1% | 60.5% | 46.8% | 66.7% | 83.4% |
| | ✓ | ✗ | 45.0 | 53.3% | 75.3% | 59.8% | 45.6% | 65.4% | **83.8%** |
| | ✓ | ✗ | 60.0 | 50.7% | 72.4% | 57.3% | 42.3% | 63.5% | 83.5% |
| | ✓ | ✗ | 90.0 | 44.6% | 65.1% | 49.9% | 35.3% | 58.9% | 81.9% |
| | ✓ | ✗ | 180.0 | 30.4% | 48.1% | 32.2% | 20.7% | 44.0% | 72.9% |
| Event + Frame | ◆ | ✓ | 20.0 | 54.9% | 74.0% | 63.2% | 50.7% | 61.3% | 85.5% |
| | ◆ | ✓ | 27.5 | 57.3% | 75.7% | 66.3% | **52.8%** | 65.8% | 86.9% |
| | ◆ | ✓ | 30.0 | 57.7% | **75.9%** | 66.8% | 52.7% | 67.2% | **87.5%** |
| | ◆ | ✓ | 36.0 | **57.8%** | 75.7% | 66.5% | 52.5% | 67.9% | 87.2% |
| | ◆ | ✓ | 45.0 | 57.2% | 75.5% | 65.4% | 51.6% | **68.2%** | **87.5%** |
| | ◆ | ✓ | 60.0 | 56.1% | 74.2% | 63.4% | 50.1% | 68.1% | 86.5% |
| | ◆ | ✓ | 90.0 | 53.7% | 72.2% | 59.5% | 47.1% | 66.2% | 85.7% |
| | ◆ | ✓ | 180.0 | 48.3% | 66.9% | 52.2% | 40.8% | 60.6% | 84.2% |
| Event + Frame | ✗ | ✓ | 20.0 | 58.0% | 76.5% | 66.4% | 52.7% | 66.2% | 86.3% |
| | ✗ | ✓ | 27.5 | 59.6% | **78.2%** | **69.6%** | **54.1%** | 69.9% | **88.0%** |
| | ✗ | ✓ | 30.0 | **60.0%** | 78.1% | 69.5% | 53.7% | **71.3%** | 87.8% |
| | ✗ | ✓ | 36.0 | 59.6% | 77.2% | 68.6% | 53.1% | 71.1% | 87.7% |
| | ✗ | ✓ | 45.0 | 59.0% | 76.7% | 67.1% | 52.1% | 71.1% | 87.8% |
| | ✗ | ✓ | 60.0 | 57.6% | 75.2% | 65.6% | 50.2% | 70.6% | 87.2% |
| | ✗ | ✓ | 90.0 | 54.8% | 72.6% | 61.9% | 46.8% | 68.8% | 86.3% |
| | ✗ | ✓ | 180.0 | 49.2% | 67.4% | 53.5% | 40.8% | 62.3% | 85.4% |
| Event + Frame | ✓ | ✓ | 20.0 | 57.4% | 78.2% | 66.6% | 51.7% | 64.9% | 83.7% |
| | ✓ | ✓ | 27.5 | 60.0% | 79.4% | 70.1% | 53.5% | 68.4% | **86.1%** |
| | ✓ | ✓ | 30.0 | 60.0% | **79.7%** | **70.8%** | **53.6%** | 69.9% | **86.1%** |
| | ✓ | ✓ | 36.0 | **60.1%** | 79.6% | **70.8%** | 53.2% | 70.3% | 85.7% |
| | ✓ | ✓ | 45.0 | 59.5% | 79.0% | 69.5% | 52.5% | 70.8% | 85.3% |
| | ✓ | ✓ | 60.0 | 58.8% | 78.5% | 69.0% | 51.1% | **71.1%** | 85.3% |
| | ✓ | ✓ | 90.0 | 56.5% | 76.5% | 65.4% | 48.2% | 70.1% | 83.8% |
| | ✓ | ✓ | 180.0 | 50.9% | 71.4% | 56.2% | 41.6% | 65.4% | 82.9% |

# D  POTENTIAL SOCIETAL IMPACT & LIMITATIONS

In this section, we discuss the potential societal impact of FlexEvent, including its positive contributions, broader implications, and known limitations. While our method offers significant advancements in event camera object detection, it is important to consider its broader consequences and areas for future improvement.

## D.1  SOCIETAL IMPACT

The development of FlexEvent introduces several positive societal benefits, particularly in safety-critical applications such as autonomous driving, robotics, and surveillance. By enhancing the ability to detect fast-moving objects in real time, our framework can improve the responsiveness and safety

Table 10: The complete results of the ablation study (*cf.* Tab. 5) of different hyperparameter configurations in the FlexEvent framework. $\tau^{car}$, $\tau^{ped}$ denotes the confidence threshold for car and pedestrian, respectively. $\tau^{iou}$ denotes the IoU threshold when filter by tracking, $L^{track}$ denotes the minimum track length. The reported results are the **mAP** scores on the validation set of *DSEC-Det* (Gehrig & Scaramuzza, 2024). The **best** and 2nd best scores of each metric from each hyperparameter configuration are highlighted in **bold** and underline, respectively.

| $\tau^{car}$ | $\tau^{ped}$ | $L^{track}$ | $\tau^{iou}$ | Frequency (Hz) | mAP | $AP_{50}$ | $AP_{75}$ | $AP_S$ | $AP_M$ | $AP_L$ |
|---|---|---|---|---|---|---|---|---|---|---|
| 0.6 | 0.3 | 10 | 0.8 | 20.0 | 56.5% | **81.3%** | 66.4% | 51.8% | 62.2% | 82.8% |
| 0.6 | 0.3 | 10 | 0.8 | 27.5 | 55.9% | 74.7% | 65.1% | 51.9% | 61.4% | 87.0% |
| 0.6 | 0.3 | 10 | 0.8 | 30.0 | 56.7% | 75.3% | 66.0% | 52.3% | 63.5% | 87.0% |
| 0.6 | 0.3 | 10 | 0.8 | 36.0 | **57.2%** | 75.9% | **67.0%** | **52.5%** | 64.8% | 86.9% |
| 0.6 | 0.3 | 10 | 0.8 | 45.0 | 57.1% | 75.7% | 66.2% | 51.5% | 66.1% | 87.2% |
| 0.6 | 0.3 | 10 | 0.8 | 60.0 | 56.7% | 75.3% | 65.7% | 50.3% | **67.4%** | **87.3%** |
| 0.6 | 0.3 | 10 | 0.8 | 90.0 | 54.5% | 73.2% | 62.3% | 47.3% | 66.3% | 86.2% |
| 0.6 | 0.3 | 10 | 0.8 | 180.0 | 49.2% | 68.2% | 54.2% | 40.8% | 62.4% | 85.5% |
| 0.6 | 0.3 | 10 | 0.6 | 20.0 | 56.7% | **80.6%** | 65.5% | 51.2% | 63.0% | 81.7% |
| 0.6 | 0.3 | 10 | 0.6 | 27.5 | 57.2% | 79.3% | 65.0% | 51.9% | 65.2% | 84.5% |
| 0.6 | 0.3 | 10 | 0.6 | 30.0 | 57.7% | 79.4% | 66.0% | **52.3%** | 66.3% | 85.0% |
| 0.6 | 0.3 | 10 | 0.6 | 36.0 | **57.9%** | 79.5% | 66.4% | 52.2% | 66.8% | 84.8% |
| 0.6 | 0.3 | 10 | 0.6 | 45.0 | 57.7% | 79.2% | 65.6% | 51.7% | 67.1% | 85.1% |
| 0.6 | 0.3 | 10 | 0.6 | 60.0 | 57.0% | 78.8% | 64.8% | 50.5% | **67.6%** | **85.3%** |
| 0.6 | 0.3 | 10 | 0.6 | 90.0 | 54.3% | 76.4% | 60.0% | 46.9% | 66.3% | 84.5% |
| 0.6 | 0.3 | 10 | 0.6 | 180.0 | 47.0% | 69.0% | 49.1% | 37.8% | 61.1% | 83.9% |
| 0.6 | 0.3 | 8 | 0.6 | 20.0 | 56.3% | 77.2% | 64.9% | 50.4% | 64.1% | 83.3% |
| 0.6 | 0.3 | 8 | 0.6 | 27.5 | 58.5% | 78.3% | 68.1% | 52.5% | 66.8% | 84.8% |
| 0.6 | 0.3 | 8 | 0.6 | 30.0 | 58.8% | 78.5% | 68.7% | 52.6% | 67.9% | 85.7% |
| 0.6 | 0.3 | 8 | 0.6 | 36.0 | **59.1%** | 78.8% | **69.0%** | **52.7%** | 68.8% | **86.5%** |
| 0.6 | 0.3 | 8 | 0.6 | 45.0 | 58.8% | 78.2% | 68.6% | 52.3% | 68.8% | 86.1% |
| 0.6 | 0.3 | 8 | 0.6 | 60.0 | 58.4% | 77.9% | 67.5% | 51.3% | **69.6%** | 85.5% |
| 0.6 | 0.3 | 8 | 0.6 | 90.0 | 56.2% | 76.6% | 64.8% | 48.5% | 68.3% | 84.7% |
| 0.6 | 0.3 | 8 | 0.6 | 180.0 | 51.2% | 71.9% | 56.3% | 42.6% | 64.2% | 82.9% |
| 0.6 | 0.3 | 6 | 0.6 | 20.0 | 57.3% | 80.0% | 65.2% | 51.2% | 65.8% | 84.1% |
| 0.6 | 0.3 | 6 | 0.6 | 27.5 | 59.4% | 81.3% | 68.5% | 53.4% | 68.8% | **85.7%** |
| 0.6 | 0.3 | 6 | 0.6 | 30.0 | 59.7% | **81.7%** | **69.0%** | **53.7%** | 69.0% | 85.3% |
| 0.6 | 0.3 | 6 | 0.6 | 36.0 | **59.9%** | 81.4% | 69.0% | 53.6% | **69.8%** | 85.7% |
| 0.6 | 0.3 | 6 | 0.6 | 45.0 | 59.3% | 80.5% | 67.9% | 52.8% | 69.4% | 85.6% |
| 0.6 | 0.3 | 6 | 0.6 | 60.0 | 58.5% | 79.6% | 67.0% | 51.5% | 69.6% | 84.9% |
| 0.6 | 0.3 | 6 | 0.6 | 90.0 | 55.7% | 77.2% | 62.8% | 48.1% | 67.9% | 84.3% |
| 0.6 | 0.3 | 6 | 0.6 | 180.0 | 48.8% | 70.6% | 50.9% | 40.8% | 61.1% | 83.4% |
| 0.6 | 0.6 | 6 | 0.6 | 20.0 | 57.4% | 78.2% | 66.6% | 51.7% | 64.9% | 83.7% |
| 0.6 | 0.6 | 6 | 0.6 | 27.5 | 60.0% | 79.4% | 70.1% | 53.5% | 68.4% | **86.1%** |
| 0.6 | 0.6 | 6 | 0.6 | 30.0 | 60.0% | **79.7%** | 70.8% | **53.6%** | 69.9% | 86.1% |
| 0.6 | 0.6 | 6 | 0.6 | 36.0 | **60.1%** | 79.6% | **70.8%** | 53.2% | 70.3% | 85.7% |
| 0.6 | 0.6 | 6 | 0.6 | 45.0 | 59.5% | 79.0% | 69.5% | 52.5% | 70.8% | 85.3% |
| 0.6 | 0.6 | 6 | 0.6 | 60.0 | 58.8% | 78.5% | 69.0% | 51.1% | **71.1%** | 85.3% |
| 0.6 | 0.6 | 6 | 0.6 | 90.0 | 56.5% | 76.5% | 65.4% | 48.2% | 70.1% | 83.8% |
| 0.6 | 0.6 | 6 | 0.6 | 180.0 | 50.9% | 71.4% | 56.2% | 41.6% | 65.4% | 82.9% |
| 0.8 | 0.8 | 6 | 0.6 | 20.0 | 56.6% | 80.7% | 65.5% | 50.8% | 65.4% | 82.6% |
| 0.8 | 0.8 | 6 | 0.6 | 27.5 | 58.7% | 81.9% | 68.9% | **52.8%** | 68.6% | 84.9% |
| 0.8 | 0.8 | 6 | 0.6 | 30.0 | **59.1%** | **82.0%** | **69.2%** | 52.7% | 69.5% | 84.8% |
| 0.8 | 0.8 | 6 | 0.6 | 36.0 | 58.9% | 81.7% | 68.8% | 52.6% | **69.9%** | 85.0% |
| 0.8 | 0.8 | 6 | 0.6 | 45.0 | 58.4% | 81.4% | 67.7% | 51.6% | 69.8% | **85.1%** |
| 0.8 | 0.8 | 6 | 0.6 | 60.0 | 57.4% | 80.1% | 66.6% | 50.2% | **69.9%** | 84.3% |
| 0.8 | 0.8 | 6 | 0.6 | 90.0 | 55.7% | 78.6% | 63.7% | 47.8% | 68.9% | 84.2% |
| 0.8 | 0.8 | 6 | 0.6 | 180.0 | 50.2% | 74.0% | 55.2% | 41.5% | 63.7% | 83.1% |

of autonomous systems operating in dynamic environments. This is especially important for avoiding collisions or responding to hazards in high-speed scenarios. For example, autonomous vehicles equipped with our approach can better detect pedestrians, cyclists, and other vehicles in real time, potentially reducing accidents and saving lives.

Additionally, the computational efficiency provided by the adaptive event-frame fusion (FlexFuser) and frequency-adaptive learning (FAL) mechanisms reduces the need for resource-intensive training processes. This contributes to the broader societal goal of making advanced AI technologies more accessible and less energy-intensive, thereby minimizing the environmental impact of large-scale AI models. Our approach could also benefit industries beyond transportation, such as robotics for healthcare, industrial automation, and public safety.

## D.2 BROADER IMPACT

The broader implications of FlexEvent include its potential to advance the field of event-based vision and enable new applications where high temporal resolution is crucial. By overcoming the limitations of conventional fixed-frequency object detection methods, our approach paves the way for more flexible, adaptable AI systems. This could lead to improvements in areas such as drone navigation, real-time video analysis for security purposes, and human-robot collaboration, where detecting fast-moving objects and adapting to changing environments are critical.

Moreover, the development of efficient and scalable detection systems like our approach can drive further innovation in resource-constrained environments, such as low-power edge devices. These advancements could make high-performance detection systems more widely available, particularly in developing regions or areas with limited access to computational resources.

However, as with any powerful technology, there is a risk of misuse. Enhanced object detection capabilities could potentially be exploited for surveillance purposes, raising privacy concerns. As event camera technology becomes more widespread, it is important to establish ethical guidelines and regulatory frameworks to ensure that these systems are used responsibly, particularly when monitoring public spaces or collecting sensitive data.

## D.3 KNOWN LIMITATIONS

While FlexEvent demonstrates significant performance improvements, there are several known limitations to our approach.

**Dependence on Event Camera Quality.** The effectiveness of our approach relies on the quality of the event camera sensor. Inconsistent or noisy event data, especially under poor lighting or extreme weather conditions, could affect detection performance. Future work could explore robustness to sensor noise and adaptation to diverse environmental conditions.

**Limited Generalization to Unseen Scenarios.** Although our approach shows strong performance across varying frequencies, it may still face challenges in completely unseen environments, where the motion dynamics and scene conditions differ significantly from the training data. Investigating methods for domain adaptation or online learning could help improve generalization to new contexts.

**Resource Requirements for High-Frequency Data.** While FlexFuser mitigates the computational cost of training on high-frequency event data, processing extremely high-frequency event streams still requires substantial computational resources during inference. This could limit the scalability on resource-constrained devices or in real-time applications with stringent latency requirements.

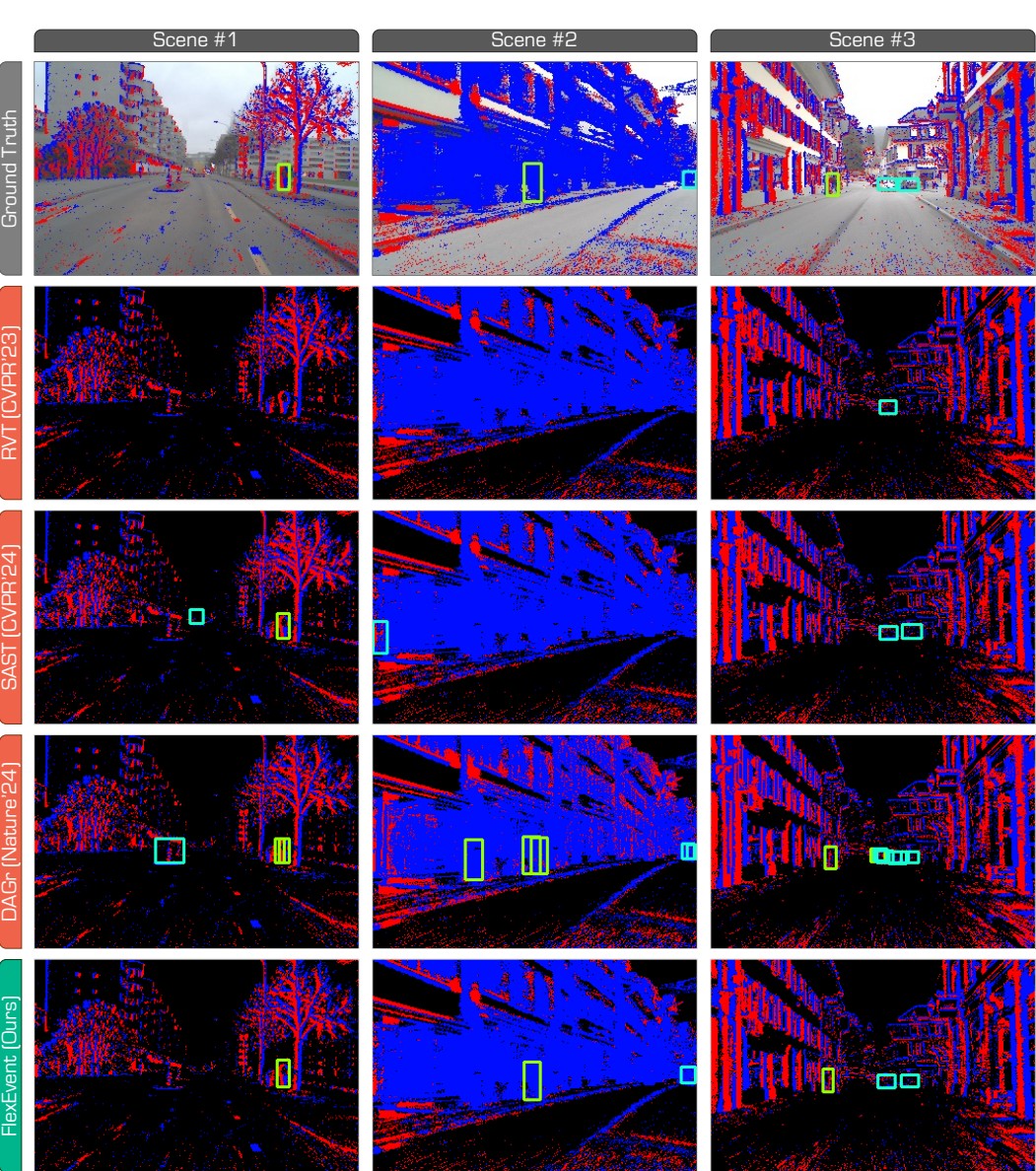

Figure 7: Additional qualitative results of state-of-the-art event camera detectors. We compare the proposed FlexEvent with RVT (Gehrig & Scaramuzza, 2023), SAST (Peng et al., 2024), and DAGr (Gehrig & Scaramuzza, 2024) on the test set of *DSEC-Det*. Best viewed in colors.

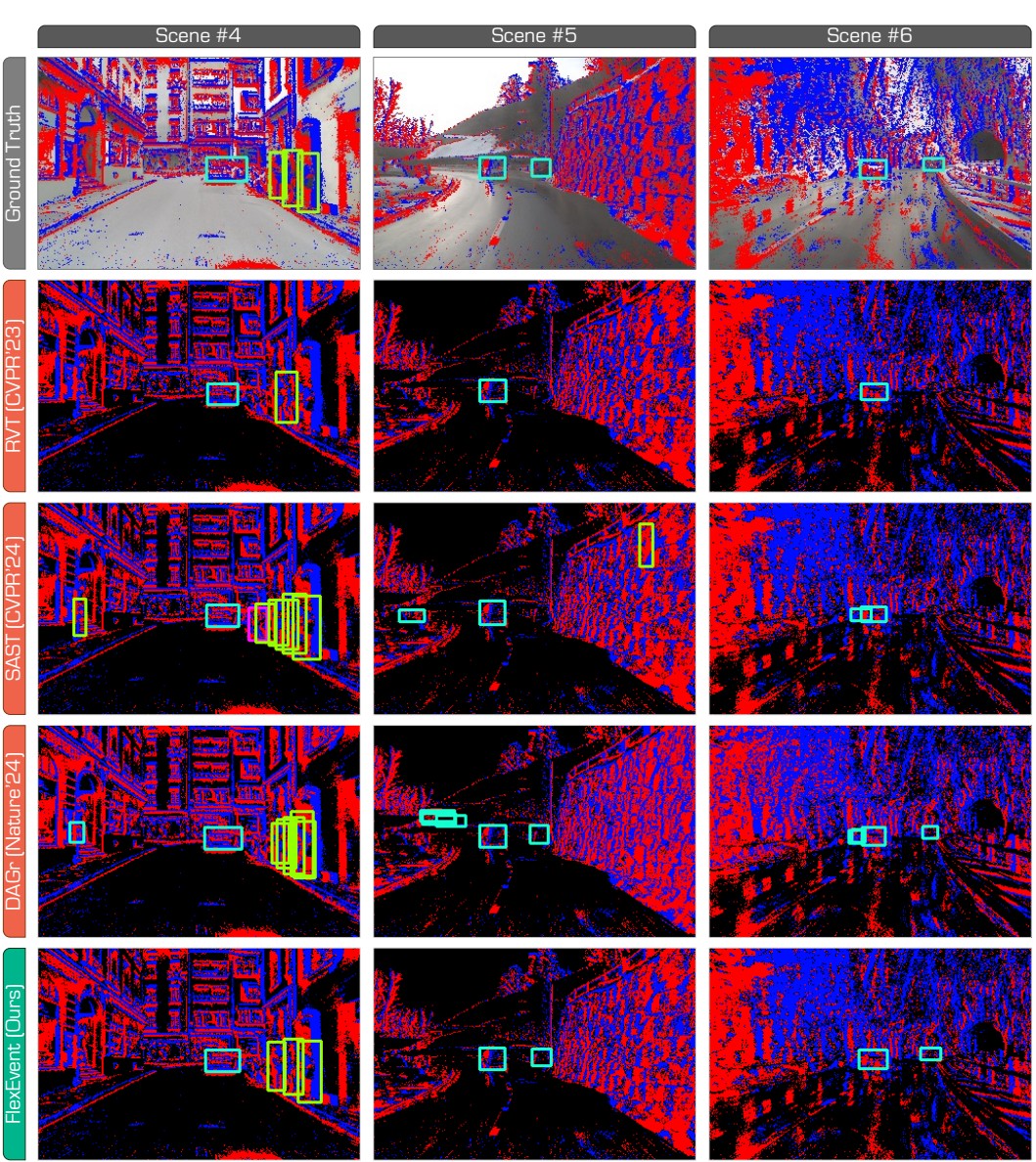

Figure 8: Additional qualitative results of state-of-the-art event camera detectors. We compare the proposed FlexEvent with RVT (Gehrig & Scaramuzza, 2023), SAST (Peng et al., 2024), and DAGr (Gehrig & Scaramuzza, 2024) on the test set of *DSEC-Det*. Best viewed in colors.

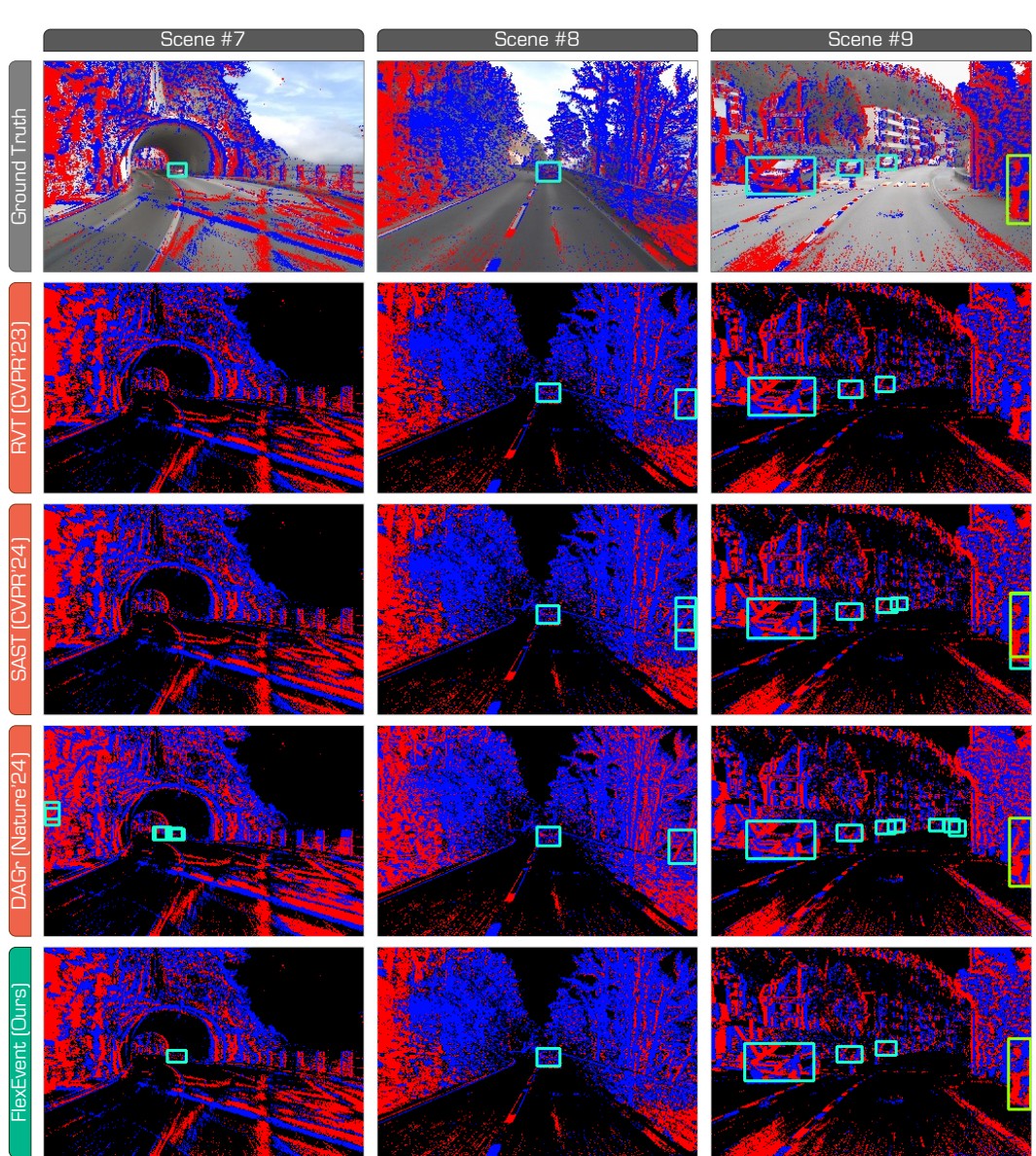

Figure 9: Additional qualitative results of state-of-the-art event camera detectors. We compare the proposed FlexEvent with RVT (Gehrig & Scaramuzza, 2023), SAST (Peng et al., 2024), and DAGr (Gehrig & Scaramuzza, 2024) on the test set of *DSEC-Det*. Best viewed in colors.

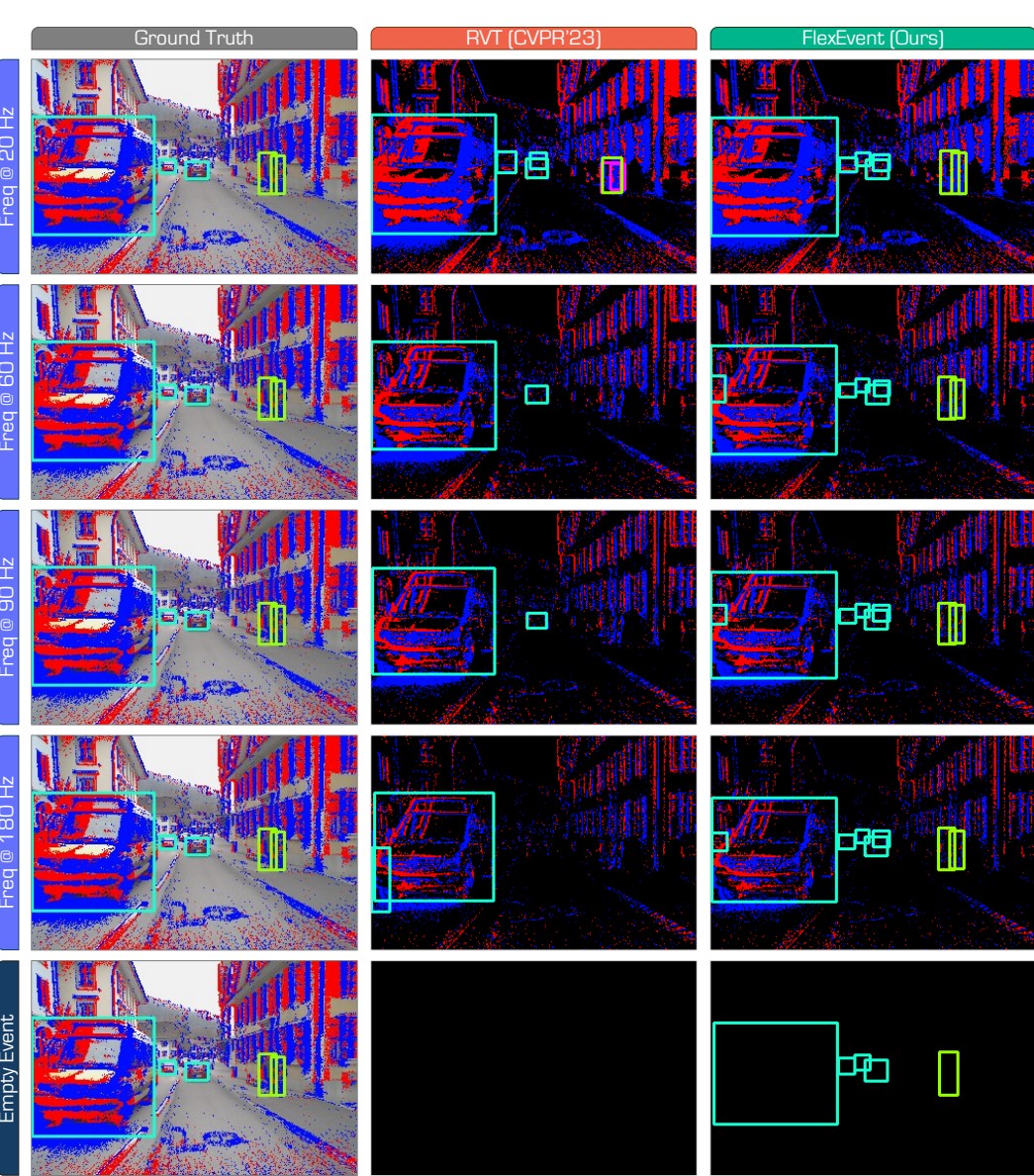

Figure 10: Additional qualitative comparisons of the RVT model (Gehrig & Scaramuzza, 2023) and the proposed FlexEvent under different event camera operation frequencies (20 Hz, 60 Hz, 90 Hz, and 180 Hz) and the empty event scenario. The experiments are conducted on the test set of *DSEC-Det*. Best viewed in colors.

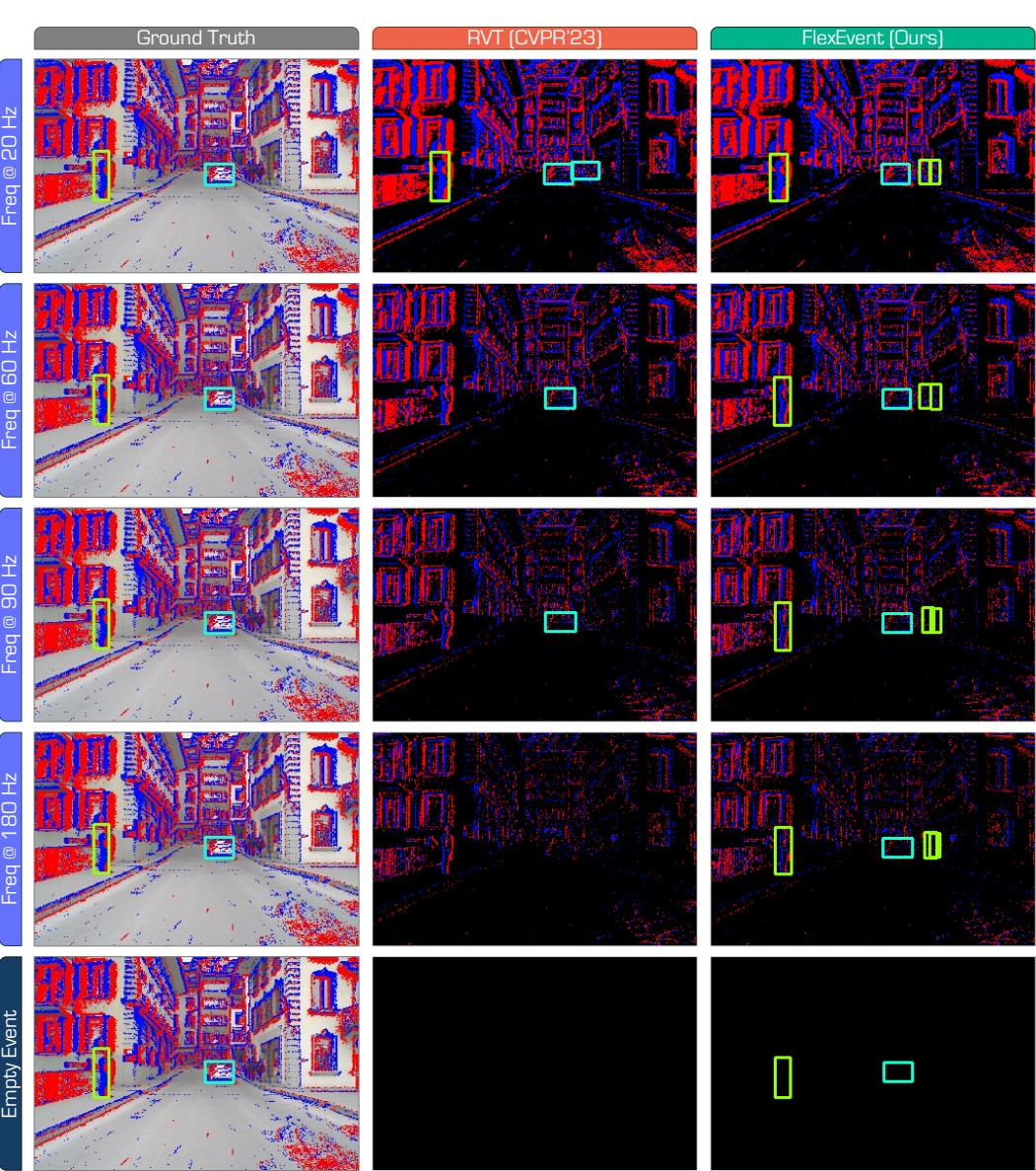

Figure 11: Additional qualitative comparisons of the RVT model (Gehrig & Scaramuzza, 2023) and the proposed FlexEvent under different event camera operation frequencies (20 Hz, 60 Hz, 90 Hz, and 180 Hz) and the empty event scenario. The experiments are conducted on the test set of *DSEC-Det*. Best viewed in colors.

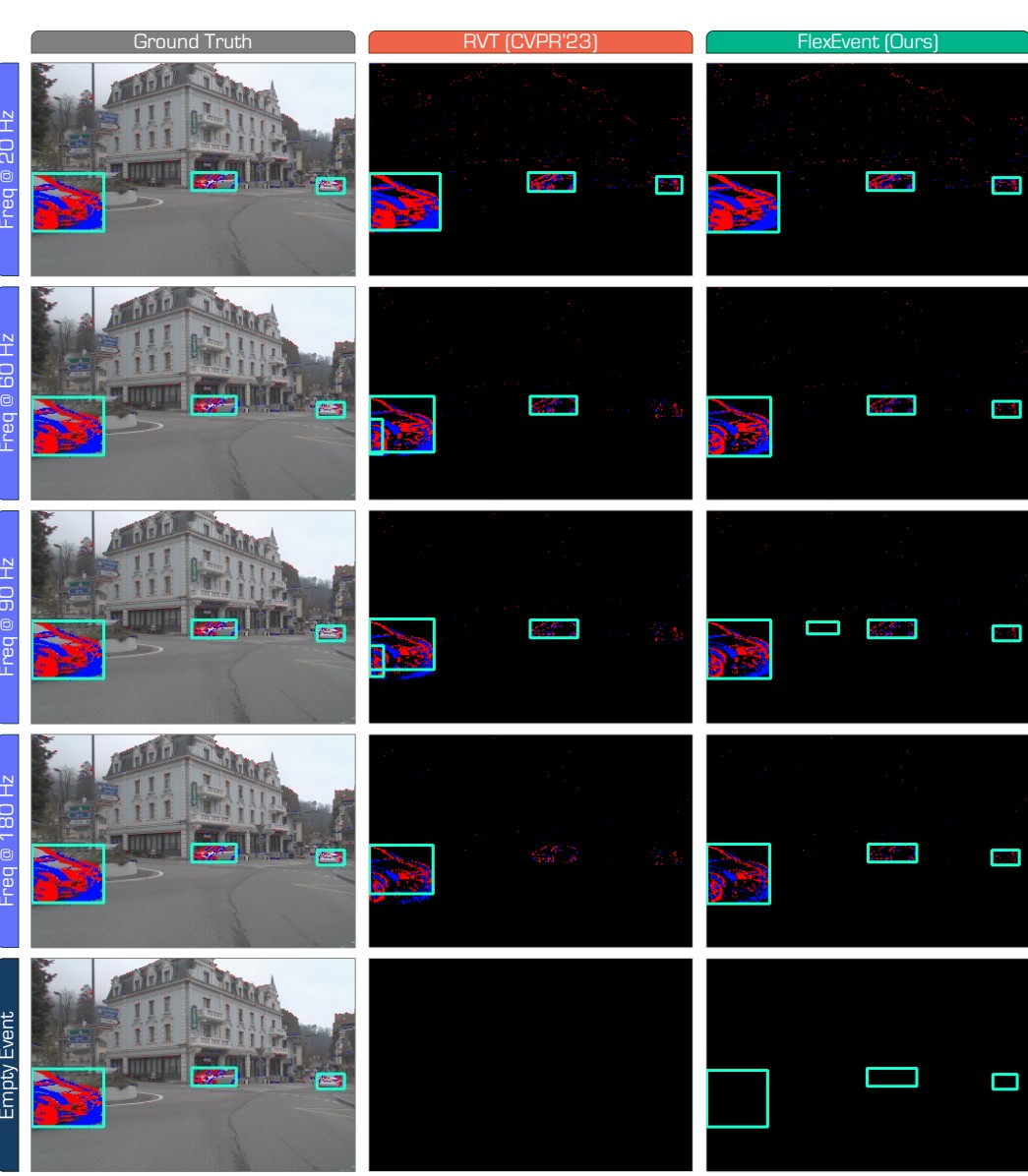

Figure 12: Additional qualitative comparisons of the RVT model (Gehrig & Scaramuzza, 2023) and the proposed FlexEvent under different event camera operation frequencies (20 Hz, 60 Hz, 90 Hz, and 180 Hz) and the empty event scenario. The experiments are conducted on the test set of *DSEC-Det*. Best viewed in colors.

# E    PUBLIC RESOURCES USED

In this section, we acknowledge the public resources used, during the course of this work.

## E.1    PUBLIC DATASETS USED

- DSEC[2] ........................................................... CC BY-SA 4.0
- DSEC-Det[3] ...................................... GNU General Public License v3.0
- DSEC-Detection[4] ............................ Creative Commons Zero v1.0 Universal
- DSEC-MOD[5] ........................................................ Unknown
- Gen 1[6] ........................ Prophesee Gen1 Automotive Detection Dataset License
- 1 Mpx[7] ....................... Prophesee 1Mpx Automotive Detection Dataset License

## E.2    PUBLIC IMPLEMENTATIONS USED

- RVT[8] ............................................................ MIT License
- SAST[9] ........................................................... MIT License
- SSM[10] ............................................................ Unknown
- LEOD[11] ........................................................... MIT License
- DAGr[12] ...................................... GNU General Public License v3.0
- RENet[13] .......................................................... Unknown

---

[2] https://dsec.ifi.uzh.ch
[3] https://github.com/uzh-rpg/dsec-det
[4] https://github.com/abhishek1411/event-rgb-fusion
[5] https://github.com/ZZY-Zhou/RENet
[6] https://www.prophesee.ai/2020/01/24/prophesee-gen1-automotive-detection-dataset
[7] https://www.prophesee.ai/2020/11/24/automotive-megapixel-event-based-dataset
[8] https://github.com/uzh-rpg/RVT
[9] https://github.com/Peterande/SAST
[10] https://github.com/uzh-rpg/ssms_event_cameras
[11] https://github.com/Wuziyi616/LEOD
[12] https://github.com/uzh-rpg/dagr
[13] https://github.com/ZZY-Zhou/RENet

