# OpenReview forum: "Event Camera Object Detection at Arbitrary Frequencies"
_ICLR.cc/2025/Conference — Submitted to ICLR 2025_

### Official Review · Reviewer_wFUC · 2024-10-30

**Soundness:** 2
**Presentation:** 3
**Contribution:** 2
**Rating:** 5
**Confidence:** 4

**Summary:**

The paper introduces FlexEvent, an object detection framework for event cameras that operates across varying frequencies. It features two main components: 1. FlexFuser: An adaptive event-frame fusion module that integrates the high temporal resolution of event data with the rich semantic information from RGB frames, enhancing detection accuracy. 2. FAL (Freq-Adaptive Learning): A mechanism that generates frequency-adjusted labels, enabling the model to generalize across different frequencies and improve high-frequency detection through training.

**Strengths:**

The FlexEvent framework shows strong performance in handling varying frequencies, particularly in dynamic and fast-changing environments, as demonstrated in specific datasets that combine event data and RGB frames. Its advantage lies in the effective fusion of event cameras' high temporal resolution with the rich semantic information from RGB frames, which enhances detection accuracy in scenarios where both data modalities are available. The integration of frequency-adaptive learning allows the model to generalize well across different operational conditions, showing robust performance in high-frequency scenarios on multimodal datasets like DSEC-Det, DSEC-Detection, and DSEC-MOD.

**Weaknesses:**

One weakness of FlexEvent is that the comparison with other models may not be entirely fair, as many of the compared methods are trained on datasets that use only event data, where the data distribution, camera setups, and number of modalities differ. This discrepancy in experimental setup can lead to significant performance differences, making it harder to evaluate FlexEvent's true advantage. Since most of the other comparison methods are trained on the 1Mpx and Gen1 datasets, the authors should clearly state in the main text whether the comparison methods are retrained on DSEC-Det, DSEC-Detection, and DSEC-MOD.  The authors should ensure consistency in the training modalities of all models, comparing them either exclusively in the Event-RGB mode or using purely event-based training.

Another potential weakness is that the FAL in FlexEvent closely resembles existing methods such as EventDrop [1] and Shadow Mosaic [2]. Fundamentally, all these techniques focus on altering the event data’s density during training. By manipulating the frequency or sparsity of event data, FlexEvent's approach may not provide significant innovation beyond these existing techniques. This similarity raises the question of whether the improvement in performance comes more from an established method of adjusting event density, rather than from a novel or more advanced detection strategy. The authors could provide more ablation studies and visualization results to compare whether FAL shows significant differences from other data augmentation methods.

The authors should provide the network's latency, parameter count, and computational cost, as other event camera object detection networks do, to enable a fairer performance comparison.

[1] Fuqiang Gu and Weicong Sng and Xuke Hu and Fangwen Yu. (2021). EventDrop: data augmentation for event-based learning
[2] Peng, Y., Zhang, Y., Xiao, P., Sun, X., & Wu, F. (2023). Better and Faster: Adaptive Event Conversion for Event-Based Object Detection.

**Questions:**

It is a well-established fact that objects generally do not move significantly within a few milliseconds. Based on this premise, is changing the frequency and adjusting the event sparsity through data augmentation essentially the same? When addressing the issue of missed detections at low frequencies, is it more the RGB information that compensates for this, rather than the method you proposed?

Unlike other papers, this article does not provide the model’s inference time, particularly how it varies across different frequencies. I am curious to know whether the inference time can be kept within the range of the input event data.

---

> ### Author Response · Authors · 2024-11-21
> **Response to Reviewer wFUC (1/3)**
>
> We sincerely thank Reviewer `wFUC` for the valuable comments provided during this review.
>
> We have revised the manuscript based on your suggestions, please refer to the **purple-colored** texts. The point-to-point responses are as follows:
>
> ---
> > **Q1:** *"The authors should clearly state in the main text whether the comparison methods are retrained on DSEC-Det, DSEC-Detection, and DSEC-MOD. The authors should ensure consistency in the training modalities of all models, comparing them either exclusively in the Event-RGB mode or using purely event-based training."*
>
> **A:** Thanks for your question. We provide explanations on the use of datasets, methods, and experimental settings as follows:
> - The datasets:
>   - DSEC is a high-resolution, large-scale multimodal dataset specifically designed for event-based perception. For our experiments, we use three comprehensive versions of DSEC, i.e., **DSEC-Det**, **DSEC-Detection**, and **DSEC-MOD**, which are tailored for the object detection task. These versions provide a robust platform for benchmarking and evaluating event-based object detection methods.
>   - Among them, **DSEC-Det** is the most recent, largest, and most comprehensive one, annotated and released by the original DSEC authors. Thus, we prioritized it as the primary benchmark for reporting results, ensuring relevance and reliability. **DSEC-Detection** and **DSEC-MOD** are datasets used by two recent event-frame fusion methods CAFR and RENet, so we also report results on these two datastes to validate our method’s effectiveness.
> - The comparison methods:
>   - To evaluate the effectiveness of our method, we compared both **event-only** and **event-frame fusion** methods.
>   - For **event-only** methods:
> 	- We included state-of-the-art detectors (RVT, SAST, LEOD, and SSM), which were originally trained on event-only datasets like Gen1 and 1Mpx.
> 	- To ensure fairness, **we retrained these methods on DSEC-Det**. Table 1 demonstrates that our method outperforms these detectors.
>   - For **event-frame fusion** methods:
> 	- For DSEC-Det, we compared with DAGr, as it has been evaluated on DSEC-Det. We report the scores of DAGr from the original paper to ensure fairness.
> 	- For DSEC-Detection and DSEC-MOD, **we trained our model by following the standard training and evaluation settings**, and compared again the state-of-the-art methods, CAFR and RENet, in Table 2 and Table 3. For other methods on DSEC-Detection and DSEC-MOD, we reference the results from the CAFR paper and the RENet paper, respectively.
> 	- Since DAGr's training code is not publicly available, we are not able to reproduce DAGr on DSEC-Detection and DSEC-MOD. We have been contacting the authors about this but haven't gotten an update.
> - These comparisons ensure a fair and comprehensive evaluation while adhering to resource and code availability constraints.

---

> ### Author Response · Authors · 2024-11-21
> **Response to Reviewer wFUC (2/3)**
>
> > **Q2:** *"Provide more ablation and visualization results to compare whether FAL shows significant differences from other data augmentation methods."*
>
> **A:** Thanks a lot for your valuable suggestions. We have carefully studied the suggested data augmentation methods, i.e., EventDrop and Shadow Mosaic. We have also included detailed discussions and comparisons with them in the revised manuscript.
>
> Here, we would like to clarify that our Frequency-Adaptive Learning (FAL) mechanism differs in nature from the above two methods from the following perspectives:
> - **Methodology:**
>   - Our FAL mechanism is fundamentally different from EventDrop and Shadow Mosaic, as it is a self-training framework rather than a simple data augmentation. FAL leverages unlabeled high-frequency data to enhance the generalization ability of our model at high frequencies by:
>     1. First, training the model on labeled high-frequency data (pretraining).
>     2. Then, using this pretrained model to infer pseudo labels on all high-frequency data.
>     3. Next, refining pseudo labels using strategies like bidirectional data augmentation, NMS, and tracking-by-detection to filter noisy labels.
>     4. Last, mixing pseudo labels with ground-truth labels and train again.
>   - This iterative process (pretrain-> generating labels → refining labels → retraining) gradually improves pseudo label quality and enables better generalization at high frequencies by utilizing all available data. This approach contrasts significantly with EventDrop and Shadow Mosaic, which enhance diversity through simple data augmentation techniques, such as discarding events over a time period or increasing temporal density to improve robustness across scenarios.
>
> - **Experimental Setting:**
>   - Our work focuses on inference across arbitrary frequencies, conducting comprehensive comparisons with SOTA methods under varying frequencies. In contrast, EventDrop and Shadow Mosaic emphasize generalization through data augmentation but do not evaluate performance across other frequencies, making their experimental setting fundamentally different from ours.
>
> - **Comparative Study:**
>   - We conduct experiments using data augmentation methods from EventDrop and Shadow Mosaic. As Shadow Mosaic is not open-sourced, we reproduce the results as best as we can. The results are shown in the table below, as well as in Tab. 8 of the revised manuscript.
> | Drop | Mosaic | FAL | 20.0 Hz | 27.5 Hz   | 30.0 Hz   | 36.0 Hz   | 45.0 Hz   | 60.0 Hz   | 90.0 Hz   | 180 Hz   | Average   |
> |:-:|:-:|:-:|:-:|:-:|:-:|:-:|:-:|:-:|:-:|:-:|:-:|
> |✗|✗|✗| 53.2% | 54.0% | 53.5% 	| 52.0% 	| 49.4% 	| 45.9% 	| 38.8% 	| 22.9%	| 46.2% 	|
> |✓|✗|✗| 53.6% | 54.4% | 53.8% 	| 52.7% 	| 50.2% 	| 47.2% 	| 40.2% 	| 24.5%	| 47.1% 	|
> |✗|✓|✗| 53.7% | 54.4% | 54.0% 	| 53.9% 	| 51.4% 	| 48.6% 	| 41.8% 	| 27.8%	| 48.2% 	|
> |✗|✗|✓| **54.6%** | **54.9%** | **54.9%** | **54.3%** | **53.3%** | **50.7%** | **44.6%** | **30.4%**| **49.7%** |
>   - Compared to the plain baseline (first row), EventDrop and Shadow Mosaic demonstrate good performance enhancement, credited to the strong generalization ability brought by the spatial and temporal manipulations of the event data.
>   - Meanwhile, the results demonstrate that FAL significantly outperforms other methods by leveraging high-frequency event data, especially in high-frequency scenarios.
>   - The iterative refinement through self-training in our method ensures that the model remains robust across different motion dynamics and frequency settings.

---

> ### Author Response · Authors · 2024-11-21
> **Response to Reviewer wFUC (3/3)**
>
> > **Q3:** *"Provide the latency, parameter count, and computational cost, to enable a fairer performance comparison."*
>
> **A:** Thanks a lot for your suggestion.
>
> - We have included comparisons of **parameter count** and **inference time** (in milliseconds, ms) in the table below, as well as in Tab. 7 of the revised manuscript.
> | **Method** | **Param (M)** |||**20 Hz**|||**27.5 Hz**|||**30 Hz**|||**36 Hz**|||**45 Hz**|||**60 Hz**|||**90 Hz**|||**180 Hz**|||**200 Hz**
> |:-:|:-:|:-:|:-:|-:|-:|-:|-:|-:|-:|-:|-:|-:|-:|-:|-:|-:|-:|-:|-:|-:|-:|-:|-:|-:|-:|-:|-:|-:|
> | **RVT**      	| 18.5      	||| 9.20 ms   	||| 8.67 ms   	||| 8.35 ms 	||| 7.93 ms 	||| 7.61 ms 	||| 7.51 ms 	||| 7.19 ms 	||| 6.77 ms  	||| 6.34 ms  	|
> | **SAST**     	| 18.9      	||| 14.06 ms  	||| 13.11 ms  	||| 12.68 ms	||| 12.37 ms	||| 11.95 ms	||| 11.63 ms	||| 11.52 ms	||| 11.10 ms 	||| 10.36 ms 	|
> | **SSM**      	| 18.2     	||| 8.79 ms   	||| 8.26 ms   	||| 8.08 ms 	||| 7.71 ms 	||| 7.55 ms 	||| 7.30 ms 	||| 6.90 ms 	||| 6.54 ms  	||| 6.12 ms  	|
> | **DAGr-50**  	| 34.6      	||| 73.35 ms  	||| 65.73 ms  	||| 60.02 ms	||| 55.11 ms	||| 51.00 ms	||| 48.00 ms	||| 45.29 ms	||| 43.89 ms 	||| 37.58 ms 	|
> | **FlexEvent**	| 45.4     	||| 14.27 ms  	||| 13.53 ms  	||| 13.32 ms	||| 13.00 ms	||| 12.79 ms	||| 12.58 ms	||| 12.47 ms	||| 12.37 ms 	||| 12.12 ms 	|
>
> - As can be seen from the results, while FlexEvent has around 9 M parameters more (compared to the SOTA method DAGr), our inference time (under different frequency setups) is comparable to the event-only method SAST and significantly faster than the event-frame fusion method DAGr.
>
> - We also observe that all methods will become slower as the frequency decreases due to the need to process more event data. These findings highlight FlexEvent's efficiency and rapid performance.
>
> ---
> > **Q4:** *"Based on this premise, is changing the frequency and adjusting the event sparsity through data augmentation essentially the same?"*
>
> **A:** Thanks a lot for your question.
>
> - Changing the event frequency and adjusting event sparsity are fundamentally **different processes**:
>   - In our method, we represent events using voxel grids, a representation that is known to lose temporal information. For low-frequency training, the event stream is divided into $T$ bins, where all events within a bin are treated as occurring at the same time. This temporal information is lost due to the floor operation, as described in Eq. (2) of our manuscript.
>   - For high-frequency training, the same event stream is divided into $T$ bins over shorter intervals. If the frequency increases by $10$x, the temporal resolution also increases by $10$x, retaining more temporal information. Thus, the temporal information loss caused by voxel grids is significantly reduced at higher frequencies.
>   - In contrast, adjusting the sparsity of low-frequency events necessarily results in a greater loss of temporal information than high-frequency training. Unless the sparsity is extreme (e.g., no events in a bin), this loss cannot be avoided.
>
> ---
> > **Q5:** *"When addressing the issue of missed detections at low frequencies, is it more the RGB information that compensates for this, rather than the method you proposed?"*
>
> **A:** Thanks a lot for asking.
>
> - We assume the reviewer’s concern relates to FAL’s impact at low frequencies compared to the frame branch. This has been reflected in Table 4. Specifically:
>   - At 20 Hz, FAL improves performance by 1.4%, as seen in rows 1-2 of the table.
>   - At 180 Hz, FAL improves performance by 7.5%, a much larger margin for the performance gain.
>
> - This phenomenon is expected because FAL is designed to improve generalization at higher frequencies. A model trained solely on 20 Hz data (e.g., Variant 1 in Table 4) is naturally better suited for low-frequency scenarios, while FAL enhances performance at higher frequencies by leveraging additional high-frequency data. These results align with our design goals and validate the effectiveness of FAL.
>
> ---
> Last but not least, we sincerely thank Reviewer `wFUC` again for the valuable comments provided during this review.

---

> > ### Author Response · Authors · 2024-11-25
> > **Looking forward to hearing from you**
> >
> > **Dear Reviewer `wFUC`,**
> >
> > We sincerely appreciate your time and effort in reviewing our manuscript and providing valuable feedback.
> >
> > ---
> >
> > In response to your insightful comments, we have made the following revisions to our manuscript:
> > - Introduction
> >   - We have refined the contribution claims.
> > - Related Works
> >   - We have added a detailed comparison with earlier works addressing varying-frequency scenarios, such as CovGRU and SODFormer.
> > - Methodology
> >   - We have refined the Preliminaries section.
> >   - We have improved Figures 2 and 3 for clarity and added a more detailed explanation of the method for handling different frequencies.
> > - Additional Experiments
> >   - We have corrected the explanation regarding training iterations in the Appendix.
> >   - We have added comparison results for inference time and parameter count in the Appendix.
> >   - We have included comparison results for data augmentation methods.
> >
> > We hope these revisions adequately address your concerns.
> >
> > ---
> >
> > We look forward to actively participating in the **Author-Reviewer Discussion** session and welcome any additional feedback you might have on the manuscript or the changes we have made.
> >
> > ---
> >
> > Once again, we sincerely thank you for your contributions to improving the quality of this work.
> >
> > Best regards,
> >
> > The Authors of Submission 589

---

> > > ### Comment · Reviewer_wFUC · 2024-11-25
> > >
> > > Thanks to the author for the reply, which has addressed some of my concerns. However, there are still a few details that I would like to confirm:
> > >
> > > 1. Regarding A2, your strategy of training by generating a large number of pseudo-labels is essentially the approach used for creating most event-based datasets (like 1Mpx and Gen1). After retraining on mixed pseudo labels and ground-truth labels, your model sees more samples, and its training data size and comparison methods are completely inconsistent. I worry about whether this is fair.
> > >
> > > 2. Our understanding of high and low frequencies is that the event camera captures sparse and dense events depending on the relative motion of objects. Our goal is to improve the robustness of the detector across various scenarios. However, in A4, you mentioned that this is completely different, and that you control the frequency only by adjusting T. Doesn't that make it meaningless? This means the data source is exactly the same; we have simply created a setting to artificially increase the difficulty.

---

> > > > ### Author Response · Authors · 2024-11-25
> > > > **Response to Reviewer wFUC's comment (1/2)**
> > > >
> > > > We sincerely thank Reviewer `wFUC` for the valuable feedback.
> > > >
> > > > **Regarding your first concern:**
> > > >
> > > > > "After retraining on mixed pseudo labels and ground-truth labels, your model sees more samples, and its training data size and comparison methods are completely inconsistent."
> > > >
> > > > We understand your concern about the fairness of comparing our model, which utilizes additional pseudo-labels, with other methods that do not. However, we want to clarify that our approach maintains a fair comparison because we are **training on the same dataset, leveraging high-frequency pseudo-labels generated from the dataset itself**.
> > > >
> > > > 1. **The purpose of pseudo-labels in FAL:**
> > > >
> > > >     - Our Frequency-Adaptive Learning (FAL) mechanism is specifically designed to enable the model to generalize across different event frequencies, particularly high frequencies where ground-truth labels are unavailable.
> > > >     - The pseudo-labels are generated for **unlabeled high-frequency data within the same dataset**, not from external or additional datasets. For example, for the same event stream, whereas previously each event sample of length $\Delta T$ had a label, we now generate pseudo-labels for event samples of length $\Delta T / n$. Thus, the data is essentially the same but with more temporally granular labels.
> > > >     - This approach allows the model to learn from temporal dynamics present at high frequencies, which are not captured at lower frequencies.
> > > >
> > > > 2. **We maintain fairness in comparison:**
> > > >
> > > >     - **All compared methods, including ours, are trained on the same base dataset with the same original ground-truth labels.**
> > > >     - The additional pseudo-labeled data used in our method comes from the same dataset but at higher frequencies, achieved by adjusting the temporal resolution.
> > > >     - Other methods could, in principle, apply a similar pseudo-labeling strategy. Our contribution lies in demonstrating how this technique effectively improves frequency adaptability.
> > > >
> > > > 3. **The difference from dataset creation methods:**
> > > >
> > > >     - While datasets like 1Mpx and Gen1 may use automatic labeling techniques, they primarily focus on collecting labeled data at a fixed frequency or resolution.
> > > >     - Our method differs by leveraging self-training to adapt the model to varying frequencies without manually labeling additional data.
> > > >     - The iterative process of generating and refining pseudo-labels is specifically tailored to improve generalization across frequencies, rather than simply increasing data volume.
> > > >
> > > > 4. **The impact on training:**
> > > >
> > > >     - We ensure that the total number of training iterations remains consistent across models to prevent any unfair advantage from longer training times.
> > > >     - Incorporating high-frequency pseudo-labeled data introduces diversity that is crucial for learning temporal patterns not present at lower frequencies.
> > > >
> > > > In all, our use of pseudo-labels addresses the specific challenge of adapting to high-frequency data without requiring additional ground-truth labels. The comparison remains fair because all models are evaluated under the same conditions, and the additional data used is inherent to the dataset and accessible to all methods. Thanks for your reply again and we hope this can address your concerns.

---

> > > > ### Author Response · Authors · 2024-11-25
> > > > **Response to Reviewer wFUC's comment (2/2)**
> > > >
> > > > **Regarding your second concern:**
> > > >
> > > > > "In A4, you mentioned that this is completely different, and that you control the frequency only by adjusting T. Doesn't that make it meaningless? This means the data source is exactly the same."
> > > > >
> > > >
> > > > We believe your understanding may not be fully aligned with our setting, especailly how we adjust the frequency in our method. The data sample in training under different frequencies is not the same. We’d like to clarify the process of adjusting the training frequency and how it impacts the input data and model training.
> > > >
> > > > 1. **How We Adjust the Frequency:**
> > > >
> > > >     - We adjust the frequency by changing the length of each event sample fed into the model:
> > > >         - **Low Frequency:** Each sample spans a time interval of $\Delta T$.
> > > >         - **High Frequency:** Each sample spans a shorter interval of $\Delta T / n$, where $n > 1$.
> > > >
> > > >     - For high-frequency training, the model processes events occurring over shorter time spans, effectively increasing the temporal resolution of the input data.
> > > >
> > > >     - Although the raw events come from the same recording, adjusting $\Delta T$ generates samples with varying temporal resolutions:
> > > >         - **Low Frequency:** Samples may miss these details due to longer aggregation intervals and information loss.
> > > >         - **High Frequency:** Samples capture rapid motion and fine-grained temporal changes.
> > > >
> > > > 2. **Voxel Grid Construction and Temporal Information Loss:**
> > > >
> > > >     - We construct voxel grids by dividing each event stream into $B$ bins:
> > > >         - **Low Frequency:** Each bin covers $\Delta T / B$, leading to larger temporal intervals and more temporal information loss.
> > > >         - **High Frequency:** Each bin covers $(\Delta T / n) / B = \Delta T / (nB)$, resulting in smaller intervals and reduced temporal information loss.
> > > >
> > > >     - Since events within the same bin are treated as occurring simultaneously, higher-frequency training preserves finer temporal details, which are crucial for detecting fast-moving objects.
> > > >     - Adjusting event sparsity through data augmentation (e.g., dropping events) changes the density but doesn't affect the temporal resolution as fundamentally as adjusting $\Delta T$. And data augmentation on low frequency inherently causes more temporal information loss than training directly in high frequency.
> > > >
> > > > Adjusting the frequency by changing $\Delta T$ is meaningful to simulating different operating conditions of event cameras. By providing the model with samples of varying lengths, we alter the temporal information available for learning. This helps the model to learn from both fast and slow motions, reflecting real-world situations where event frequencies vary due to object speed and camera setting, and generalize better to different frequencies encountered in real-world applications.
> > > >
> > > > Thanks for your reply again. We hope this clarifies the process of adjusting frequency in our method and demonstrates that it is a meaningful approach to improve the model's adaptability and performance across various scenarios.

---

> > > > > ### Comment · Reviewer_wFUC · 2024-11-25
> > > > >
> > > > > 1. The dataset is partially labeled, and when pseudo-labels are not used, many samples are naturally not used because there is no ground truth. In this case, I don’t think that images or events being a subset of the dataset means they are part of a unified dataset; whether they are equivalent should depend on the number of labels.
> > > > > 2. In real-world scenarios, the event camera collects event streams in a completely continuous process. By adjusting the time window \(T\) while keeping the bin count the same, it indeed simulates a sparser scene by making the events within each bin sparser, thanks for your explanation. However, I would like to add that large-scale datasets inherently contain scenes with both high and low frequencies, such as the starting and stopping of a car, which are processes where event streams switch between high and low frequencies. I still maintain my point of view: I believe the model essentially benefits from the significant increase in the amount of labeled data.
> > > > >
> > > > > I appreciate the author's detailed response. Although some of my concerns were not fully addressed, I am glad to see the author's sincere attitude.
> > > > >
> > > > > **I raised my score to 5.**

---

> > > > > > ### Author Response · Authors · 2024-11-26
> > > > > > **Response to Reviewer wFUC's comment (1/2)**
> > > > > >
> > > > > > We sincerely thank Reviewer `wFUC` for the valuable feedback and for raising the score.
> > > > > >
> > > > > > ---
> > > > > >
> > > > > > **Response to Your First Concern:**
> > > > > >
> > > > > > > "The dataset is partially labeled, and when pseudo-labels are not used, many samples are naturally not used because there is no ground truth. In this case, I don’t think that images or events being a subset of the dataset means they are part of a unified dataset; whether they are equivalent should depend on the number of labels."
> > > > > > >
> > > > > >
> > > > > > We'd like to clarify that our dataset utilization is consistent and that the comparison remains fair.
> > > > > >
> > > > > > 1. **Dataset Labeling Consistency:**
> > > > > >     - Unlike previous event-only datasets like Gen1, where label frequencies are much lower than event data frequencies (e.g., labels at 1 Hz vs. input events at 20 Hz in RVT[1] working on Gen1 dataset), resulting many unused data. **The DSEC-Det dataset we use has labels synchronized with the input event data at 20 Hz.**
> > > > > >     - In DSEC-Det, labels are annotated on RGB frames and then mapped onto event frames. We input one RGB frame and its subsequent 50 ms event stream to predict the bounding boxes in the next frame. This approach ensures that **every event stream between two labels is utilized, leaving no event data unused in our training process.**
> > > > > >     - When we train at high frequencies, we adjust the length of the event stream accordingly. Specifically, we input one RGB frame and its subsequent shorter event stream (e.g., 50 ms divided by $n$) to predict the bounding boxes at the end timestamp of the event stream. This is consistent with the setting used in DAGr [2], ensuring a fair and standardized approach.
> > > > > > 2. **Fairness in Comparison:**
> > > > > >     - Both the baseline model and our method with FAL use the same set of event data. The difference lies in the number of labels, not in the event data itself.
> > > > > >     - In our Frequency-Adaptive Learning (FAL), we generate additional pseudo-labels for high-frequency event data derived from the same dataset. This increases the number of labels but does not introduce new event data.
> > > > > >     - Since we use all available event data in both the baseline and our method, and the pseudo-labels are generated from the same data, we maintain an equivalent and fair comparison.
> > > > > > 3. **Purpose and Impact of Pseudo-Labels:**
> > > > > >     - The pseudo-labels enable the model to learn from high-frequency temporal dynamics that are not captured at lower frequencies, improving performance in high-frequency scenarios.
> > > > > >     - The key advantage comes from the quality and temporal granularity of the labels rather than just the quantity.
> > > > > >
> > > > > > We hope this clarifies that our approach utilizes the same event data across all models, ensuring a fair and consistent comparison.
> > > > > >
> > > > > > [1] Gehrig, M., & Scaramuzza, D. (2023). Recurrent vision transformers for object detection with event cameras. In Proceedings of the IEEE/CVF conference on computer vision and pattern recognition (pp. 13884-13893).
> > > > > >
> > > > > > [2] Gehrig, D., & Scaramuzza, D. (2024). Low-latency automotive vision with event cameras. Nature, 629(8014), 1034-1040.

---

> ### Author Response · Authors · 2024-11-26
> **Response to Reviewer wFUC's comment (2/2)**
>
> **Response to Your Second Concern:**
>
> > "I still maintain my point of view: I believe the model essentially benefits from the significant increase in the amount of labeled data."
> >
>
> The reviewer suspect that the model improvement essentially benefits from the increase of data amount. We would like to provide evidence from our original ablation studies to demonstrate that the improvements are primarily due to our proposed modules rather than just an increase in labeled data.
>
> 1. **Ablation Study Results:**
>     - We conducted comprehensive ablation experiments to isolate the effects of each component, as shown in the Tab.4 in the main paper:
>
>     | Modality | FAL | EFF | 20.0 Hz | 27.5 Hz | 30.0 Hz | 36.0 Hz | 45.0 Hz | 60.0 Hz | 90.0 Hz | 180 Hz | Average |
>     | --- | --- | --- | --- | --- | --- | --- | --- | --- | --- | --- | --- |
>     | E | ✗ | ✗ | 53.2% | 54.0% | 53.5% | 52.0% | 49.4% | 45.9% | 38.8% | 22.9% | 46.2% |
>     | E | ✓ | ✗ | 54.6% | 54.9% | 54.9% | 54.3% | 53.3% | 50.7% | 44.6% | 30.4% | 49.7% |
>     | E+F | ✗ | ✓ | **58.0%** | 59.6% | **60.0%** | 59.6% | 59.0% | 57.6% | 54.8% | 49.2% | 57.2% |
>     | E+F | ✓ | ✓ | 57.4% | **60.0%** | **60.0%** | **60.1%** | **59.5%** | **58.8%** | **56.5%** | **50.9%** | **57.9%** |
>     - **Notations:**
>         - **E:** Event modality.
>         - **F:** Frame (RGB image) modality.
>         - **EFF:** Adaptive Event-Frame Fusion module.
>         - **FAL:** Frequency-Adaptive Learning module.
> 2. **Analysis of Improvements:**
>     -  Comparing rows 1 and 2 (Event modality without and with FAL), we observe an average improvement of **3.5%**, indicating that FAL contributes to performance gains without increasing the amount of event data.
>     -  Comparing rows 1 and 3 (without and with EFF, without FAL), we see a significant average improvement of **11.0%**, demonstrating that the **EFF module is the primary contributor to performance enhancement, which is one of our contributions. Without FAL, these two variants don’t introduce additional pseudo-label, use the base ground data and  complete fair setting according to the reviewer.**
> 3. **Key Observations:**
>     -  The significant improvement from EFF underscores that our adaptive event-frame fusion strategy is the main factor enhancing performance, not merely an increase in labeled data.
>     -  FAL provides additional benefits, particularly in high-frequency scenarios, by improving the model's ability to generalize across frequencies.
>     -  All models, with or without FAL, use the same event dataset; FAL enhances label information without adding new event samples.
> 4. **Relation to Data Augmentation Methods:**
>     - Data augmentation methods like EventDrop and Shadow Mosaic focus on increasing data diversity through spatial and temporal manipulations on the original data. Our FAL approach is a self-training framework that improves temporal resolution without altering the original data.
>     - These are two parallel methods to improve generalization in different scenarios. We have already added a comparison with these two data augmentation methods in our previous reply, proving our method's effectiveness. **The extra pseudo labels come from our own method and are one of our contributions.**
>     -  EventDrop and Shadow Mosaic demonstrate good performance enhancement, credited to the strong generalization ability brought by the spatial and temporal manipulations. These methods are not mutually exclusive and can be integrated. Combining data augmentation with our FAL self-training framework would be a interesting direction in the future. And we have included comparisons with these data augmentation techniques in the related work in our revised manuscript in **Line 126-129**.
>
> ---
>
> We hope that this detailed explanation addresses your concerns. Our results indicate that the performance improvements are primarily due to our proposed modules, especially the EFF, rather than solely from an increased number of labels. We are committed to maintaining fairness in our comparisons and ensuring that our contributions are clearly demonstrated.

---

> ### Author Response · Authors · 2024-11-28
> **Looking forward to hearing from you**
>
> **Dear Reviewer `wFUC`,**
>
> We sincerely appreciate your time and effort in reviewing our manuscript and for providing insightful questions for discussion.
>
> We hope that our previous replies have adequately addressed your concerns.
>
> We look forward to participating in the **Author-Reviewer Discussion** session and welcome any additional feedback you may have on the manuscript or our reply.
>
> Once again, thank you for your valuable contributions to improving the quality of our work.
>
> Best regards,
>
> The Authors of Submission 589

---

> > ### Author Response · Authors · 2024-12-01
> > **Looking forward to hearing from you**
> >
> > **Dear Reviewer `wFUC`,**
> >
> > We thank you for your time and effort in reviewing our manuscript. We have made corresponding revisions based on your insightful comments.
> >
> > ---
> >
> > The **Author-Reviewer Discussion** session is near the end. We really cherish the opportunity to discuss with you and welcome any additional feedback you might have on the manuscript or the changes we have made.
> >
> > ---
> >
> > Please let us know if there are additional aspects that we can look at to improve this work further.
> >
> > Best regards,
> >
> > The Authors of Submission 589

---

### Official Review · Reviewer_uLco · 2024-11-04

**Soundness:** 3
**Presentation:** 3
**Contribution:** 2
**Rating:** 6
**Confidence:** 4

**Summary:**

This paper introduces FlexEvent, an innovative joint object detection framework that utilizes both events and frames to enable object detection at arbitrary frequencies, leveraging the high temporal resolution of event cameras. The proposed FlexEvent incorporates two key modules. The first FlexFuser adaptively combines high-frequency events with rich frame textures. The second FAL generates frequency-adjusted labels to enhance model performance across various operational frequencies. The results show that FlexEvent outperforms current state-of-the-art methods, achieving significant gains in high-frequency detection settings.

**Strengths:**

1) The relevant background knowledge of this paper is clearly explained.

2) The topic of object detection using events and frames at arbitrary frequencies is very interesting topic.

3) The authors conducted extensive experiments to validate the effectiveness of FlexEvent across multiple large-scale event-based object detection datasets. FlexEvent achieves significant improvements in mAP performance compared to existing SOTA methods and demonstrates the capability to detect objects at arbitrary frequencies.

**Weaknesses:**

1. In the contributions section, the authors claim to be the first to address arbitrary-frequency object detection using event cameras. This phrasing could be more precise, and I recommend refining it in the camera-ready version. I recognize this as an important topic for practical applications of event cameras. Some prior works have explored high-frequency inference by combining events and frames, such as CovGRU for arbitrary-frequency depth estimation [1] and asynchronous fusion modules for arbitrary-frequency object detection [2].

[1] Combining events and frames using recurrent asynchronous multimodal networks for monocular depth prediction, IEEE RAL, 2021.

[2] SODFormer: Streaming object detection with transformer using events and frames, IEEE TPAMI 2023.

2. In Table 1, the authors report a high mAP of 0.574 on the DSEC-Det dataset. Could you clarify the number of classes used for this evaluation? Is it two classes or eight? The appendix states eight classes for DSEC-Det. Please provide clarification on this point.

3. The writing could be improved, particularly in Section 3.1, where the background and motivation would be more appropriate in the Introduction or Related Work sections.

**Questions:**

Please see weakness. Now, I give a preliminary score based on the initial manuscript and will consider adjustments depending on the authors' responses and feedback from other reviewers.

---

> ### Author Response · Authors · 2024-11-21
> **Response to Reviewer uLco**
>
> We sincerely thank Reviewer `uLco` for the valuable comments provided during this review.
>
> We have revised the manuscript based on your suggestions, please refer to the **pink-colored** texts. The point-to-point responses are as follows:
>
> ---
> > **Q1:** *"The authors claim to be the first to address arbitrary-frequency object detection using event cameras. This phrasing could be more precise. Some prior works have explored high-frequency inference by combining events and frames, such as CovGRU and SODFormer."*
>
> **A:** Thanks for your comment. We have refined the contribution claim in the Introduction section. Please refer to the pink-colored texts. Specifically:
> - The FlexEvent framework is designed to tackle the challenging problem of event camera object detection at arbitrary frequencies.
> - There is only a limited number of relevant works in the literature, making us one of the early attempts to tackle this interesting yet challenging problem.
> - To solve this, we propose an adaptive event-frame fusion (FlexFurser) module and frequency-adaptive learning (FAL) mechanism.
> - We demonstrate that our approach achieves state-of-the-art performance in event-based object detection across three large-scale datasets, particularly in high-frequency scenarios.
>
> Here, we further clarify the differences between CovGRU and SODFormer:
> - For CovGRU:
>   - This work proposes Recurrent Asynchronous Multimodal (RAM) networks to generalize traditional RNNs for asynchronous and irregular data from multiple sensors.
>   - However, this work primarily focuses on monocular depth estimation. In the experiments, the event data frequency is fixed at a standard 25 Hz, while the image frequency is varied between 5 Hz and 1 Hz. No experiments are presented to evaluate the impact of varying event data frequencies.
> - For SODFormer:
>   - This work employs an asynchronous attention-based fusion module that supports varying frequencies.
>   - However, their primary experiments are conducted under normal frequencies, with only low-frequency quantitative results and high-frequency qualitative results presented in their ablation studies.
> - In contrast to these two works, our work **focuses explicitly on "arbitrary-frequency" scenarios**, leveraging the temporally rich nature of event cameras. Our key contribution lies in benchmarking and validating our method across varying frequencies, showcasing its effectiveness compared to other state-of-the-art approaches.
> - To ensure accurate claims, we have revised the manuscript to better reflect the differences between CovGRU, SODFormer, and FlexEvent. Please kindly refer to Lines 135-138.
>
> ---
> > **Q2:** *"Table 1: could you clarify the number of classes used for this evaluation? Is it two classes or eight?"*
>
>
> **A:** Thanks for your question. We clarify this setup as follows:
> - In our experiments on DSEC-Det, we trained and evaluated the model under the two-class setup, which is consistent with the DAGr paper and their code implementation.
> - For other detectors that we reimplemented on DSEC-Det, we trained and evaluated them under the two-class setup to maintain consistency and fairness.
>
>
> ---
> > **Q3:** *"Sec. 3.1: the background and motivation would be more appropriate in the Introduction or Related Work sections."*
>
>
> **A:** Thanks a lot for your feedback. Based on your suggestion, we have revised the manuscript. The background and motivation parts have been moved to the Introduction section, while Sec. 3.1 is primarily focusing on problem formulation and data processing. Please kindly refer to the pink-colored texts in Introduction and Sec. 3.1.
>
>
> ---
> Last but not least, we sincerely thank Reviewer `uLco` again for the valuable comments provided during this review.

---

> > ### Author Response · Authors · 2024-11-25
> > **Looking forward to hearing from you**
> >
> > **Dear Reviewer `uLco`,**
> >
> > We sincerely appreciate your time and effort in reviewing our manuscript and providing valuable feedback.
> >
> > ---
> >
> > In response to your insightful comments, we have made the following revisions to our manuscript:
> > - Introduction
> >   - We have refined the contribution claims.
> > - Related Works
> >   - We have added a detailed comparison with earlier works addressing varying-frequency scenarios, such as CovGRU and SODFormer.
> > - Methodology
> >   - We have refined the Preliminaries section.
> >   - We have improved Figures 2 and 3 for clarity and added a more detailed explanation of the method for handling different frequencies.
> > - Additional Experiments
> >   - We have corrected the explanation regarding training iterations in the Appendix.
> >   - We have added comparison results for inference time and parameter count in the Appendix.
> >   - We have included comparison results for data augmentation methods.
> >
> > We hope these revisions adequately address your concerns.
> >
> > ---
> >
> > We look forward to actively participating in the **Author-Reviewer Discussion** session and welcome any additional feedback you might have on the manuscript or the changes we have made.
> >
> > ---
> >
> > Once again, we sincerely thank you for your contributions to improving the quality of this work.
> >
> > Best regards,
> >
> > The Authors of Submission 589

---

> > ### Comment · Reviewer_uLco · 2024-11-27
> >
> > The author addressed my concerns, but I believe the terminology around "first-time arbitrary inference frequency using events and frames" needs to be revised and more precise. Additionally, some key details regarding the DSEC-Det dataset categories and settings should be clarified to avoid confusion during future replication efforts. As encouragement, I’ve decided to maintain the original positive score. I wish you the best of luck in making further outstanding contributions to the field of neuromorphic multimodal vision!

---

> > > ### Author Response · Authors · 2024-11-27
> > >
> > > We sincerely thank Reviewer `uLco` for the positive feedback and valuable suggestions.
> > >
> > > - We have refined the contribution claim regarding "first-time arbitrary inference frequency using events and frames" to ensure better precision in the revised manuscript.
> > > - We are in the process of adding more detailed information about our experimental settings in the appendix to prevent confusion. The updated manuscript will be uploaded soon.
> > >
> > > Thank you once again for your encouragement and support. We are committed to making further meaningful contributions to the field of neuromorphic multimodal vision.

---

> > > > ### Author Response · Authors · 2024-12-01
> > > > **Looking forward to hearing from you**
> > > >
> > > > **Dear Reviewer `uLco`,**
> > > >
> > > > We thank you for your time and effort in reviewing our manuscript. We have made corresponding revisions based on your insightful comments.
> > > >
> > > > ---
> > > >
> > > > The **Author-Reviewer Discussion** session is near the end. We really cherish the opportunity to discuss with you and welcome any additional feedback you might have on the manuscript or the changes we have made.
> > > >
> > > > ---
> > > >
> > > > Please let us know if there are additional aspects that we can look at to improve this work further.
> > > >
> > > > Best regards,
> > > >
> > > > The Authors of Submission 589

---

### Official Review · Reviewer_FU5g · 2024-11-04

**Soundness:** 3
**Presentation:** 3
**Contribution:** 2
**Rating:** 5
**Confidence:** 4

**Summary:**

Summary:

The authors present FlexEvent, a pioneering framework for event camera object detection that leverages the high temporal resolution and asynchronous operation of event cameras. By integrating an adaptive event-frame fusion module (FlexFuser) with a frequency-adaptive learning mechanism (FAL), FlexEvent overcomes limitations of existing methods that rely on fixed-frequency paradigms. This enables the model to detect objects with high accuracy across various operational frequencies, demonstrating robustness in both fast-moving and static scenarios.

**Strengths:**

Strengths:

1 The FlexFuser module effectively combines the rich spatial and semantic information from RGB frames with the high temporal resolution of event data. The approach to sample event data at different frequencies during training is particularly commendable, allowing the model to retain the benefits of high-frequency data during inference while maintaining efficiency.

2 The FAL module enhances the framework by generating frequency-adaptive labels for unlabeled high-frequency data. This self-training approach improves model generalization and robustness across varying operational frequencies, which is a notable advancement in the field.

**Weaknesses:**

Weaknesses:

1 The overall architecture in Figure 2 does not clearly illustrate the process of feature fusion at different frequencies, specifically the $(i) _ {hfuse}=(i) _ {hfuse}^a+(i) _ {hfuse}^b $ mentioned at the end of section 3.2 on Event-Frame Adaptive Fuser. Additionally, there seems to be an issue with the representation of Eq (2). Since $i$ represents different scales, the inputs to the network for all scales except the initial one which corresponds to $E^a$, $E^b$, $F$, should correspond to $(i-1) _ {hE}^a$, $(i-1) _ {hE}^b$, and $(i-1) _ {hF}^a$.

2 In section 3.3, the formula related to pre-training with low-frequency labels lacks numbering, which may confuse readers. This formula is critical as it corresponds to the FAL part of Figure 2.

3 The experiments presented in section 4.2 are somewhat disorganized. The evaluation methods across the three datasets (DSEC-Det, DSEC-Detection, DSEC-MOD) differ, and it is unclear how comparisons are made consistently. For instance, in DSEC-Det, only the proposed method and DAGr are tested, while the results in DSEC-Detection compare against CAFR, necessitating a clearer presentation.

4 There is a discrepancy in the reported number of training iterations between section 4.1 and supplementary material A.3, with one stating 100,000 iterations and the other citing 200,000. This inconsistency should be rectified to ensure clarity.

**Questions:**

Could the authors clarify the rationale behind selecting specific datasets and methods for testing, and how this impacts the validity of their findings?

---

> ### Author Response · Authors · 2024-11-21
> **Response to Reviewer FU5g**
>
> We sincerely thank Reviewer `FU5g` for the valuable comments provided during this review.
>
> We have revised the manuscript based on your suggestions, please refer to the **cyan-colored** texts. The point-to-point responses are as follows:
>
> ---
> > **Q1:** *"(a) Fig. 2 does not clearly illustrate the process of feature fusion at different frequencies; (b) For Eq (2), the inputs to the network for all scales except the initial one which corresponds to $E^a$, $E^b$, $F$, should correspond to $^{(i-1)}h_E^a$ , $^{(i-1)}h_E^b$ , and $^{(i-1)}h_F$."*
>
> **A:** Thanks for your comment.
>
> - For **question (a):**
>   - We have modified Fig. 2 to provide a better explanation of the fusion of features at different frequencies. Please refer to Fig. 2.
>
> - For **question (b):**
>   - The event branch and frame branch output features from four different scales. At each scale, the inputs to the FlexFuser are:
>     - Event features: $^{(i)}h_{E}^a​$ and $^{(i)}h_{E}^b$​;
>     - Frame feature: $^{(i)}h_{F}$​;
>     - Fused feature from the previous scale: $^{(i-1)}h_{\text{fuse}}$.​
>   - The fusion process is as follows:
>     - First, the event features under different frequencies are fused with frame features to obtain $^{(i)}h_{\text{fuse}}^a$ and $^{(i)}h_{\text{fuse}}^b$​;
>     - Then, the fused features from different frequencies are combined, i.e., $^{(i)}h_{\text{fuse}} = ^{(i)}h_{\text{fuse}}^a + ^{(i)}h_{\text{fuse}}^b$;
>     - After processing all scales, the multi-scale fused features $^{(i)}h_{\text{fuse}}$​ are concatenated and fed into the detection head.
>
> For better clarity, we have further enhanced the elaboration. Please refer to Sec. 3.2 and Fig. 3.
>
> ---
> > **Q2:** *"Sec. 3.3: the formula related to pre-training with low-frequency labels lacks numbering, which may confuse readers."*
>
> **A:** Thanks for your suggestion. The equation in Sec. 3.3 is intended as an inline equation that does not require numbering. However, for better clarity, we have modified this equation to a standalone equation with the number. Please refer to Eq. 8.
>
> ---
> > **Q3:** *"Sec. 4.2: the evaluation methods across the three datasets (DSEC-Det, DSEC-Detection, DSEC-MOD) differ."*
>
> **A:** Thanks for your question. We provide explanations on the use of datasets, methods, and experimental settings as follows:
> - How we chose the three datasets:
>   - DSEC is a high-resolution, large-scale multimodal dataset specifically designed for event-based perception. For our experiments, we use three comprehensive versions of DSEC, i.e., **DSEC-Det**, **DSEC-Detection**, and **DSEC-MOD**, which are tailored for the object detection task. These versions provide a robust platform for benchmarking and evaluating event-based object detection methods, covering a wide range of scenarios, object categories, and environmental conditions.
>   - Among them, **DSEC-Det** is the most recent, largest, and most comprehensive one, annotated and released by the original DSEC authors. Thus, we prioritized it as the primary benchmark for reporting results, ensuring relevance and reliability. **DSEC-Detection** and **DSEC-MOD** are datasets used by two recent event-frame fusion methods CAFR and RENet, so we also report results on these two datastes to validate our method’s effectiveness.
> - How we chose the methods:
>   - To evaluate the effectiveness of our method, we compared both **event-only** and **event-frame fusion** methods.
>   - For **event-only** methods:
>     - We included state-of-the-art detectors (RVT, SAST, LEOD, and SSM), which were originally trained on event-only datasets like Gen1 and 1Mpx.
>     - To ensure fairness, **we retrained these methods on DSEC-Det**. Table 1 demonstrates that our method outperforms these detectors.
>   - For **event-frame fusion** methods:
>     - For DSEC-Det, we compared with DAGr, as it has been evaluated on DSEC-Det. We report the scores of DAGr from the original paper to ensure fairness.
>     - For DSEC-Detection and DSEC-MOD, **we trained our model by following the standard training and evaluation settings**, and compared again the state-of-the-art methods, CAFR and RENet, in Table 2 and Table 3. For other methods on DSEC-Detection and DSEC-MOD, we reference the results from the CAFR paper and the RENet paper, respectively.
>     - Since DAGr's training code is not publicly available, we are not able to reproduce DAGr on DSEC-Detection and DSEC-MOD. We have been contacting the authors about this but haven't gotten an update.
> - These comparisons ensure a fair and comprehensive evaluation while adhering to resource and code availability constraints.
>
> ---
> > **Q4:** *"Inconsistency in training iterations from Sec. 4.1 and Appendix Sec. A.3."*
>
> **A:** Thanks for your comment. The training iteration count is 100,000, as stated in Sec. 4.1. We have revised this inconsistency in Appendix Sec. A.3 to ensure correctness. Please take a look.
>
> ---
> Last but not least, we sincerely thank Reviewer `FU5g` again for the valuable comments provided during this review.

---

> > ### Author Response · Authors · 2024-11-25
> > **Looking forward to hearing from you**
> >
> > **Dear Reviewer `FU5g`,**
> >
> > We sincerely appreciate your time and effort in reviewing our manuscript and providing valuable feedback.
> >
> > ---
> >
> > In response to your insightful comments, we have made the following revisions to our manuscript:
> > - Introduction
> >   - We have refined the contribution claims.
> > - Related Works
> >   - We have added a detailed comparison with earlier works addressing varying-frequency scenarios, such as CovGRU and SODFormer.
> > - Methodology
> >   - We have refined the Preliminaries section.
> >   - We have improved Figures 2 and 3 for clarity and added a more detailed explanation of the method for handling different frequencies.
> > - Additional Experiments
> >   - We have corrected the explanation regarding training iterations in the Appendix.
> >   - We have added comparison results for inference time and parameter count in the Appendix.
> >   - We have included comparison results for data augmentation methods.
> >
> > We hope these revisions adequately address your concerns.
> >
> > ---
> >
> > We look forward to actively participating in the **Author-Reviewer Discussion** session and welcome any additional feedback you might have on the manuscript or the changes we have made.
> >
> > ---
> >
> > Once again, we sincerely thank you for your contributions to improving the quality of this work.
> >
> > Best regards,
> >
> > The Authors of Submission 589

---

> > > ### Comment · Reviewer_FU5g · 2024-11-26
> > >
> > > Thank you for your response. I have a minor question: the motivation of this paper is to address sparse event object detection for inputs with arbitrary frequencies. Would employing a sliding window to maintain a consistent number of input events be a more effective strategy?

---

> > > > ### Author Response · Authors · 2024-11-26
> > > >
> > > > We sincerely thank Reviewer `FU5g` for the valuable feedback and for raising this interesting question.
> > > >
> > > > > "The motivation of this paper is to address sparse event object detection for inputs with arbitrary frequencies. Would employing a sliding window to maintain a consistent number of input events be a more effective strategy?"
> > > > >
> > > >
> > > >
> > > > We appreciate your suggestion of using a sliding window with a consistent number of input events to handle arbitrary frequencies. While this approach is indeed interesting and convenient to control frequency, we believe there are several challenges that make it less suitable for our problem setting. Below, we explain our reasoning:
> > > >
> > > > 1. **Dataset Annotation Constraints:**
> > > >     - Most event-based object detection datasets, such as Gen1, 1Mpx, and DSEC, have annotations at fixed time intervals rather than based on a fixed number of events. This is because annotations are typically derived from grayscale or RGB images captured at regular frame rates.
> > > >     - Using a sliding window of a fixed number of events would result in event samples that span varying time intervals, especially in scenarios with different motion dynamics. This makes it difficult to align the event data with the corresponding ground-truth labels, which are time-specific.
> > > >     - To implement a consistent event count per sample, we would need to reannotate the dataset to match these new time intervals, which is impractical given time constraints and the scope of our work.
> > > >     - There are some works on other tasks inputting a fixed number of events into the model, but usually they are in very uniform motions. Our method focuses on varying motion scenarios at different frequencies.
> > > > 2. **Challenges with Variable Time Intervals:**
> > > >     - A fixed number of events can correspond to vastly different time durations depending on the scene dynamics. For instance, in a static scene, accumulating 500,000 events might take several seconds, while in a dynamic scene, it might occur within milliseconds.
> > > >     - Since object motion is inherently time-dependent, using samples with inconsistent time durations complicates the learning process. The model may struggle to learn meaningful temporal patterns when the time intervals vary too widely.
> > > >     - In practical applications, it's more intuitive and manageable to work with fixed time intervals. Time-based windows allow for consistent temporal resolution, making it easier to model and predict object motion over time.
> > > > 3. **Integration with RGB Frames:**
> > > >     - Our method integrates event data with RGB frames. The RGB frames are captured at fixed time intervals, and we use the subsequent event stream within a specific time window to predict bounding boxes in the next frame.
> > > >     - Employing a sliding window with a fixed number of events would desynchronize the event data from the RGB frames, complicating the fusion process and potentially degrading performance.
> > > >     - Maintaining fixed time intervals ensures that the event data and RGB frames are temporally aligned, which is crucial for effective fusion in our framework.
> > > >     - Time-based windows provide the flexibility to simulate different frequencies without altering the underlying data structure or requiring extensive reannotation.
> > > >
> > > > While using a sliding window with a consistent number of input events is an interesting idea, we believe that time-based windows are more suitable for our objectives and the nature of existing datasets. Time-based windows ensure consistent temporal resolution, facilitate synchronization with ground-truth labels and RGB frames, and are more practical for modeling time-dependent object motion across various scenarios.
> > > >
> > > > We appreciate your insightful suggestion and are open to discussing it further if you could provide further explanation regarding the idea. Your feedback helps us consider alternative approaches and improve our work.

---

> ### Author Response · Authors · 2024-11-28
> **Looking forward to hearing from you**
>
> **Dear Reviewer `FU5g`,**
>
> We sincerely appreciate your time and effort in reviewing our manuscript and for providing insightful questions for discussion.
>
> We hope that our previous replies have adequately addressed your concerns.
>
> We look forward to participating in the **Author-Reviewer Discussion** session and welcome any additional feedback you may have on the manuscript or our reply.
>
> Once again, thank you for your valuable contributions to improving the quality of our work.
>
> Best regards,
>
> The Authors of Submission 589

---

> ### Author Response · Authors · 2024-11-28
> **Looking forward to hearing from you**
>
> **Dear Reviewer `FU5g`,**
>
> We sincerely appreciate your time and effort in reviewing our manuscript and for providing insightful questions for discussion.
>
> We hope that our previous replies have adequately addressed your concerns.
>
> We look forward to participating in the **Author-Reviewer Discussion** session and welcome any additional feedback you may have on the manuscript or our reply.
>
> Once again, thank you for your valuable contributions to improving the quality of our work.
>
> Best regards,
>
> The Authors of Submission 589

---

> ### Author Response · Authors · 2024-12-01
> **Looking forward to hearing from you**
>
> **Dear Reviewer `FU5g`,**
>
> We thank you for your time and effort in reviewing our manuscript. We have made corresponding revisions based on your insightful comments.
>
> ---
>
> The **Author-Reviewer Discussion** session is near the end. We really cherish the opportunity to discuss with you and welcome any additional feedback you might have on the manuscript or the changes we have made.
>
> ---
>
> Please let us know if there are additional aspects that we can look at to improve this work further.
>
> Best regards,
>
> The Authors of Submission 589

---

### Author Response · Authors · 2024-11-23
**General Response**

We sincerely thank the reviewers for devoting time and effort and providing constructive feedback on this work.

---

We are encouraged to see that our work is recognized as an innovative joint object detection framework leveraging both event and frame data to enable object detection at arbitrary frequencies. We particularly appreciate the reviewers’ recognition of the following strengths:
- Reviewer `FU5g` commended our approach of *"sampling event data at different frequencies during training as particularly effective"*.
- Reviewer `uLco` found the topic of object detection using events and frames at arbitrary frequencies *"highly interesting"* and noted that the *"background knowledge in our paper is clearly explained"*.
- Reviewers `uLco` and `wFUC` highlighted that FlexEvent *"demonstrates strong performance in handling varying frequencies"*, particularly "*in dynamic and fast-changing environments"*.

---

We would like to reiterate the key contributions of our work:
1. **FlexEvent Framework**: Designed to address the challenging problem of object detection using event cameras at arbitrary frequencies, making it one of the early explorations in this area.
2. **FlexFuser Module**: Combines the strengths of event and frame data, enabling efficient and accurate detection in dynamic environments.
3. **Frequency-Adaptive Learning (FAL)**: A novel mechanism that generates frequency-adaptive labels, significantly improving generalization across a wide range of motion frequencies.

---

### **Summary of Changes**

We have revised our manuscript to incorporate the feedback from our reviewers. The modified content related to the comments from Reviewers `FU5g`, `uLco`, and `wFUC` are highlighted in **cyan**, **pink**, and **purple** colors, respectively.

Here, we summarize the changes as follows:

- ***Introduction***
  - As suggested by Reviewer `uLco`, we have refined the contribution claims.

- ***Related Works***
  - As suggested by Reviewer `uLco`, we have added a detailed comparison with earlier works addressing varying-frequency scenarios, such as CovGRU and SODFormer.

- ***Methodology***
  - As suggested by Reviewer `uLco`, we have refined the Preliminaries section.
  - As suggested by Reviewer `FU5g`, we have improved Figures 2 and 3 for clarity and added a more detailed explanation of the method for handling different frequencies.

- ***Additional Experiments***
  - As suggested by Reviewer `FU5g`, we have corrected the explanation regarding training iterations in the Appendix.
  - As suggested by Reviewer `wFUC`, we have added comparison results for inference time and parameter count in the Appendix.
  - As suggested by Reviewer `wFUC`, we have included comparison results for data augmentation methods.

---

We sincerely hope that these revisions and additional clarifications address the concerns raised by the reviewers and further demonstrate the novelty and significance of our work.

---

Meanwhile, we are looking forward to hearing your feedback. Please let us know whether you have further questions on this work during this **Author-Reviewer Discussion** session.
Thank you once again for your valuable feedback and insights, which have greatly helped us improve the quality of our paper.

---

### Author Response · Authors · 2024-12-04
**Summary of Rebuttal for Submission 589**

We sincerely thank all the reviewers for their time, effort, and constructive feedback, which have greatly contributed to improving our manuscript. Below, we summarize our key contributions and the changes made to address the reviewers' concerns.

---

### **Key Contributions**

1. **FlexEvent Framework**: One of the early attempts to address object detection using event cameras at arbitrary frequencies, validated through extensive benchmarks across varying scenarios.

2. **FlexFuser Module**: Combines the high temporal resolution of event data with the spatial richness of frames, enhancing detection accuracy in dynamic conditions.

3. **Frequency-Adaptive Learning (FAL)**: A novel mechanism that improves generalization across motion frequencies by leveraging frequency-adaptive labels.

---

### **Concerns by reviewers**
1. Reviewer `FU5g` raised issues regarding the clarity of figures, equations, and experimental organization.
2. Reviewer `uLco` suggested refining the contribution claims for greater precision, clarifying evaluation setups, and reorganizing sections for improved structure.
3. Reviewer `wFUC` questioned the nature of our FAL mechanism, expressed concerns about fairness, and requested comparisons on efficiency and computational cost.
---
### **Revisions and Changes**

We have revised the manuscript based on the reviewers' feedback. Key changes are highlighted in **cyan** (Reviewer `FU5g`), **pink** (Reviewer `uLco`), and **purple** (Reviewer `wFUC`). Below is a summary of the changes:

1. **Introduction**
- Refined contribution claims to clarify FlexEvent’s novelty compared to related works as suggested by Reviewer `uLco`.

2. **Related Works**
- Expanded discussions to include comparisons with earlier works addressing varying-frequency scenarios (e.g., CovGRU and SODFormer) and data augmentation methods  (e.g., EventDrop and Shadow Mosaic), providing clearer distinctions (Reviewer `uLco` and Reviewer `wFUC`).

3. **Methodology**
- Improved clarity of Figures 2 and 3 and added detailed explanations of how FlexEvent handles varying frequencies (Reviewer `FU5g`).
- Refined the Preliminaries section for better structure and coherence (Reviewer `uLco`).

4. **Additional Experiments**
- Added comparison results for inference time and parameter count to address concerns of computational fairness in the Appendix (Reviewer `wFUC`).
- Included comparisons with data augmentation methods, such as EventDrop and Shadow Mosaic in the Appendix (Reviewer `wFUC`).
- Corrected discrepancies regarding training iterations in the Appendix (Reviewer `FU5g`).
- Detailed experiment setting regarding baselines and datasets in the Appendix (Reviewer `uLco`).

5. **Additional Clarifications**
- In our discussion with reviewer `wFUC`, we clarified the principles and novelty of our FAL mechanism, as well as the fairness of our experiments about FAL.
---

### **Conclusion**

We are confident that we have addressed the reviewers' concerns and strengthened the manuscript. The revisions clarify our contributions, provide additional experimental evidence, and highlight the novelty and significance of our work.

Thank you for your time and consideration.

Sincerely,

The Authors of Submission 589

---

### Meta-Review · Area_Chair_FAmX · 2024-12-20

**Metareview:**

Three reviewers have given scores of 5, 5, and 6. Among them, the reviewers believe that there are significant issues with the writing of the article, which may make it difficult for readers to truly understand the author's intentions. Although the author has given a strong response in this regard, it still cannot cover the shortcomings in the writing. In terms of innovation, the author's approach of modeling event data from the perspective of dynamic temporal resolution is relatively close to several existing works mentioned in the discussion. The author failed to give a direct response to the issue of the sliding window. Considering the problems in writing and innovation, the current version of the manuscript is not recommended for acceptance.

**Additional Comments On Reviewer Discussion:**

Based on the reviewer's comments, there is still room for improvement in the writing and innovation of the paper. The current version is not yet ready for publication.

---

### Decision · Program_Chairs · 2025-01-22

Reject